# Boundary-to-Region Supervision for Offline Safe Reinforcement Learning

**Huikang Su**$^{\diamond *}$  **Dengyun Peng**$^{\clubsuit *}$  **Zifeng Zhuang**$^{\spadesuit}$  **Yuhan Liu**$^{\diamond}$
**Qiguang Chen**$^{\clubsuit}$  **Donglin Wang**$^{\spadesuit}$  **Qinghe Liu**$^{\diamond \dagger}$

$^{\diamond}$Harbin Institute of Technology, Weihai,    $^{\clubsuit}$ Harbin Institute of Technology, Harbin
$^{\spadesuit}$ Westlake University, Hangzhou

suhuikang123@gmail.com,  dypeng@ir.hit.edu.cn
qingheliu@hitwh.edu.cn

## Abstract

Offline safe reinforcement learning aims to learn policies that satisfy predefined safety constraints from static datasets. Existing sequence-model-based methods condition action generation on symmetric input tokens for return-to-go and cost-to-go, neglecting their intrinsic asymmetry: return-to-go (RTG) serves as a flexible performance target, while cost-to-go (CTG) should represent a rigid safety boundary. This symmetric conditioning leads to unreliable constraint satisfaction, especially when encountering out-of-distribution cost trajectories. To address this, we propose Boundary-to-Region (B2R), a framework that enables asymmetric conditioning through cost signal realignment . B2R redefines CTG as a boundary constraint under a fixed safety budget, unifying the cost distribution of all feasible trajectories while preserving reward structures. Combined with rotary positional embeddings , it enhances exploration within the safe region. Experimental results show that B2R satisfies safety constraints in 35 out of 38 safety-critical tasks while achieving superior reward performance over baseline methods. This work highlights the limitations of symmetric token conditioning and establishes a new theoretical and practical approach for applying sequence models to safe RL. Our code is available at https://github.com/HuikangSu/B2R.

## 1 Introduction

Offline reinforcement learning (RL) enables policy learning from static datasets without risky online interactions [28, 7], a critical capability for safety-sensitive applications such as autonomous driving [21, 23, 37], robotics [4, 5], and industrial control systems [36]. While conventional offline RL focuses on maximizing rewards under distributional shift [17, 38, 24], real-world deployments often demand adherence to safety constraints [9, 3]. This necessitates offline safe RL, which seeks policies that maximize cumulative rewards while ensuring expected costs remain below predefined thresholds.

Recent advances in Reinforcement Learning via Supervised Learning (RvS), exemplified by the Decision Transformer (DT) [2], have shown promise by autoregressively generating actions conditioned on historical states, actions, and return-to-go (RTG) signals. However, extending DT to safe RL reveals a fundamental limitation: existing methods naively apply symmetric conditioning mechanisms to both RTG and cost-to-go (CTG), overlooking their inherent asymmetry. Specifically, RTG serves as a flexible performance target to pursue, while CTG represents a rigid safety budget to enforce—a distinction that existing approaches fail to capture.

---

$^{*}$Equal contribution
$^{\dagger}$Corresponding Author

39th Conference on Neural Information Processing Systems (NeurIPS 2025).

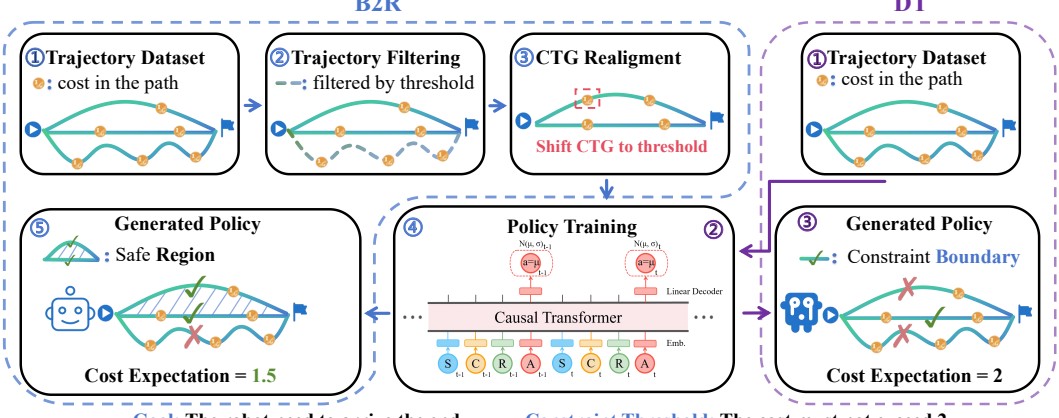

Figure 1: Overview of the B2R framework compared to DT methods. DT approaches rely on boundary-aligned trajectories whose costs happen to match the constraint threshold, making it difficult to supervise diverse safe behaviors and often resulting in unsafe, high-cost actions. To address this, the B2R pipeline introduces **Trajectory Filtering** to remove unsafe samples and **CTG Realignment** to align all remaining trajectories with the deployment-time cost threshold. This transforms sparse boundary supervision into consistent training over a broader **Safe Region**, reducing expected cost (1.5). In contrast, DT methods lack such filtering and alignment, frequently generating actions beyond the constraint, leading to higher expected cost (2.0), as shown in the right subfigure.

This oversight leads to unreliable constraint satisfaction [1], particularly when policies encounter cost trajectories outside the training distribution. Methods like Constrained Decision Transformer (CDT) [26] treat RTG and CTG as equivalent input tokens, conflating the orthogonal objectives of reward maximization and safety assurance. To address this, we propose Boundary-to-Region (B2R), a framework that introduces asymmetric conditioning through CTG realignment. Our core innovation lies in unifying feasible trajectories under a fixed safety budget by redistributing cost signals while preserving reward structure. This redefines CTG as a boundary constraint rather than a variable target, decoupling safety guarantees from reward optimization. Combined with trajectory filtering and rotary positional embeddings [31], B2R enables comprehensive exploration of the safe action space while maintaining strict cost adherence. Figure 1 illustrates how B2R broadens supervision beyond narrow constraint-aligned trajectories, enabling stable learning over the entire safe region.

Experiments on 38 safety-critical tasks demonstrate B2R's effectiveness: it satisfies safety constraints in 35 environments while achieving competitive rewards [25]. Our findings underscore the necessity of abandoning symmetric token conditioning when applying sequence models to safe RL. **The contributions of this work are threefold:**

**1. Problem Identification**: We identify and formalize a fundamental symmetry fallacy in existing RvS methods [2, 26], where the flexible nature of rewards and the rigid nature of costs are improperly treated as symmetric signals.

**2. Methodology and Validation**: We propose region-level supervision, a new paradigm for offline safe RL. We then introduce B2R, a logically coherent framework designed to instantiate this paradigm, demonstrating how components like trajectory filtering and CTG realignment form a mutually reinforcing system, not a collection of ad-hoc tweaks.

**3. Theoretical Foundation**:We provide initial theoretical analysis of B2R's safety compliance [1] under simplified assumptions and empirically validate its superiority in balancing reward and safety across 38 diverse tasks.

## 2 Related Work

### 2.1 Offline Reinforcement Learning and RvS

Offline RL learns policies from static datasets [20], with methods addressing distributional shift by constraining policies near the behavior policy, such as BCQ [8], CQL [17], BPPO[39], and BEAR

[16]. While these methods leverage value-based objectives [33] to mitigate policy deviations, they often face challenges such as the "deadly triad" [32].

Reinforcement Learning via Supervised Learning [6] reframes reinforcement learning as a supervised learning problem [30, 27], where policies are conditioned on signals such as goals or rewards. Decision Transformer [2] uses a transformer to autoregressively generate actions conditioned on states and returns-to-go. Reinformer [40] improves upon DT by incorporating a return-maximizing objective into sequence models, guiding action selection without relying on pre-specified RTG. ConDT [15] introduces an enhanced contrastive loss to structure return-dependent representations, leading to significant performance gains.

## 2.2 Offline Safe Reinforcement Learning

Offline Safe RL combines safe RL [9, 12] and offline RL by enforcing safety constraints in static datasets to ensure reliable policy learning. Early methods, such as Lagrangian optimization [34] and distribution correction [19], incorporate cost constraints during training but are less flexible with diverse datasets. A key focus in Offline Safe RL is enhancing dataset quality to address safety challenges. OASIS [35] uses a conditional diffusion model to reshape datasets for improved safety compliance, while partitioning-based approaches [10] classify trajectories into desirable and undesirable subsets to avoid unsafe behaviors. Constrained Decision Transformer [26] highlights the importance of trajectory data by leveraging sequential modeling to enforce safety constraints. FAWAC [13] uses a learned feasibility critic to down-weight unsafe actions during policy optimization in offline RL. LSPC [14] constrains policy learning in a latent trajectory space inferred from both reward and safety signals.

While prior methods have made significant strides, B2R is distinctly positioned by addressing the fundamental challenge of sparse and symmetric supervision in sequence-based offline safe RL. Unlike approaches such as CDT [26], which are limited to sparse **boundary supervision** by conditioning on specific cost-to-go values, B2R introduces **region-wide supervision**. By realigning the costs of all safe trajectories to a unified boundary, it learns from a much denser and more diverse set of behaviors. Furthermore, B2R differs from filtering-based methods like TraC [10], which classify and often discard large portions of the dataset. Instead of merely selecting safe trajectories, B2R's core contribution is to **transform** them, retaining their behavioral diversity to create a more robust policy. Thus, B2R's novelty lies in its principled approach to reshaping the supervision signal itself to resolve the underlying symmetry fallacy.

# 3 Preliminaries

## 3.1 Offline Safe Reinforcement Learning

We formalize the safe reinforcement learning problem using the Constrained Markov Decision Process (CMDP) framework [1]. A CMDP $\mathcal{M}$ is defined as a tuple $(\mathcal{S}, \mathcal{A}, P, r, c, \mu_0)$, where $\mathcal{S}$ and $\mathcal{A}$ denote the state and action spaces, $P : \mathcal{S} \times \mathcal{A} \times \mathcal{S} \to [0, 1]$ is the transition probability function, $r : \mathcal{S} \times \mathcal{A} \times \mathcal{S} \to \mathbb{R}$ is the reward function, $c : \mathcal{S} \times \mathcal{A} \times \mathcal{S} \to [0, C_{\max}]$ is the cost function bounded by a known constant $C_{\max} \geq 0$, and $\mu_0$ is the initial state distribution over $\mathcal{S}$.

We define the cumulative reward and cost of a trajectory $\tau = \{(s_t, a_t, r_t, c_t)\}_{t=0}^{H}$ as $R(\tau) = \sum_{t=0}^{H-1} r(s_t, a_t, s_{t+1})$ and $C(\tau) = \sum_{t=0}^{H-1} c(s_t, a_t, s_{t+1})$, respectively, where $H$ is the horizon length. A policy $\pi : \mathcal{S} \times \mathcal{A} \to [0, 1]$ is considered feasible if its expected cumulative cost does not exceed the threshold $\kappa \in [0, +\infty)$, i.e., $\mathbb{E}[C(\tau)] \leq \kappa$.

The goal of offline safe reinforcement learning is to learn a feasible policy $\pi$ that maximizes the expected return while satisfying a cost constraint:

$$\pi^* = \arg\max_{\pi} \mathbb{E}_{\tau \sim \pi}[R(\tau)] \quad \text{subject to} \quad \mathbb{E}_{\tau \sim \pi}[C(\tau)] \leq \kappa. \tag{1}$$

In the offline setting, the agent has no access to environment interaction and must instead learn from a static dataset $\mathcal{D} = \{\tau_i\}_{i=1}^{N}$ collected by unknown or suboptimal behavior policies. This introduces significant challenges such as distributional shift and limited coverage, making it difficult to simultaneously achieve high return and strict safety compliance.

## 3.2 Reinforcement Learning via Supervised Learning

To address distributional shift in offline reinforcement learning, *Reinforcement Learning via Supervised Learning* [6] reformulates policy learning as conditional sequence modeling. Instead of estimating value functions or applying dynamic programming, RvS learns it as conditional distributions $\pi(a_t|x_t)$, where $x_t$ is a user-specified signal like return, goal, or trajectory-level attributes.

A representative approach in the RvS paradigm is the **Decision Transformer** [2], which treats offline reinforcement learning as conditional sequence modeling. In safety-aware variants, each trajectory is represented as a sequence of tokens $(\hat{R}_t, \hat{C}_t, s_t, a_t)$, where *return-to-go* $\hat{R}_t = \sum_{k=t}^{H-1} r_k$ and *cost-to-go* $\hat{C}_t = \sum_{k=t}^{H-1} c_k$ denote future cumulative reward and cost. The model is trained to predict actions autoregressively from prior context via behavior cloning:

$$\mathcal{L}_{\text{BC}}(\theta) = \mathbb{E}_{\tau^{(n)} \sim \mathcal{D}} \left[ -\log \pi_\theta \left( a_t^{(n)} \mid \hat{R}_{t-K:t}^{(n)}, \hat{C}_{t-K:t}^{(n)}, s_{t-K:t}^{(n)}, a_{t-K:t-1}^{(n)} \right) \right] \qquad (2)$$

In this formulation, $\tau^{(n)} \sim \mathcal{D}$ denotes a trajectory from the offline dataset, the parameter $K$ defines the length of the context window, and $\theta$ denotes the parameters of policy model. The model predicts the next action $a_t^{(n)}$ given a fixed-length context window of return, cost, state, and action tokens from steps $t - K$ to $t$. During deployment, the model generates actions autoregressively given the initial condition tokens $(\hat{R}_0, \hat{C}_0)$ and the observed state sequence.

# 4 Method

To address the symmetry fallacy identified in the introduction, this section details the Boundary-to-Region framework. B2R is not a collection of ad-hoc components but a coherent system designed to implement our proposed region-wide supervision paradigm. This system consists of three interlocking parts: (1) trajectory filtering to define the safe region, (2) CTG realignment to create a dense and uniform supervision signal, and (3) RoPE to preserve the temporal dynamics of this signal. The following subsections will elaborate on the motivation and implementation of each component within this unified system.

## 4.1 Limitations of Symmetric Token Conditioning

In CMDP formulations, the optimization objective (Eq. 1) reveals a fundamental asymmetry: reward signals rank feasible policies, while cost signals define whether a policy is feasible at all. This asymmetry is not merely a modeling artifact—it is essential to the semantics of safe decision making.

However, modern DT-style sequence models do not reflect this asymmetry. Instead, they encode both reward and cost as RTG and CTG tokens, treating them symmetrically to condition action prediction. At each timestep, these tokens are computed from the trajectory's future cumulative reward and cost, and are used as conditioning signals during training, as defined by the loss in Eq. 2. At deployment time, the model is queried with user-specified initial values $\hat{R}_0$ and $\hat{C}_0$, representing the desired return and the target cost budget. **This failure to reflect the asymmetry** between reward and cost, gives rise to two key challenges in safe offline policy deployment.

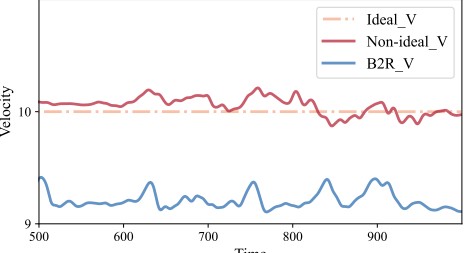

Figure 2: Velocity profiles in a simplified MetaDrive scenario. Training on boundary-aligned trajectories results in unstable behavior and frequent violations (*Non-ideal_V*), while B2R achieves smooth, constraint-compliant control.

**The first is the difficulty of token selection itself.** In DT-style models, both $\hat{R}_0$ and $\hat{C}_0$ are treated symmetrically as conditioning signals, but they serve very different purposes: $\hat{R}_0$ sets a performance target, while $\hat{C}_0$ enforces a safety limit. Choosing these values requires balancing ambition and feasibility. A large return target $\hat{R}_0$ encourages high-return behavior, but overly optimistic values may push the policy into out-of-distribution (OOD) regions not supported by the offline data. At the same time, determining a compatible cost token $\hat{C}_0$ that supports the desired return while satisfying safety constraints is particularly difficult, as the appropriate

trade-off is often unknown and hard to infer from data. Notably, trajectories with cost close to the safety threshold are sparse, and high-return behaviors may occur at lower cost levels.

The second challenge remains even when the initial CTG is fixed to a known deployment constraint, typically $\hat{C}_0 = \kappa$. In this case, the model is expected to imitate high-reward behavior under constraint threshold, but the dataset may contain few trajectories whose cumulative cost is close to $\kappa$, limiting the quality of supervision available at training time. Moreover, while reward prediction errors degrade performance, **cost modeling errors can cause constraint violations.** This risk is especially high near the constraint threshold, where small errors can lead to unsafe, hard-to-generalize behavior.

Figure 2 illustrates this issue more clearly in a simplified MetaDrive environment. The agent is rewarded for higher speeds but penalized when velocity exceeds 10. This structure creates a distinction within the safe region itself: behaviors at the boundary (v=10) are less robust and relatively riskier than those with a safety margin (v<10). The *Non-ideal_V* curve, trained only on boundary-aligned (v=10) trajectories, learns a brittle policy that constantly overcorrects, failing to respect this nuance. In contrast, **B2R**, by learning from diverse behaviors deep within the safe region, develops a robust control policy that successfully maintains this crucial safety margin.

## 4.2   From Boundary Supervisionto Region-Wide Supervision

To address these challenges, we introduce a consistent training strategy that aligns all trajectories with an initial CTG token $\hat{C}_0 = \kappa$, while broadening supervision beyond the narrow set around boundary-aligned examples. B2R accomplishes this by CTG realignment in the dataset to expose the full spectrum of safe behaviors. We now formalize the supervision distinction central to this design:

**Definition 1** (Constraint Boundary- vs. Safe Region-Conditional Supervision). At time step $t$, the learner receives a cost-to-go token $\hat{C}_t \in (0, \kappa]$. Given a suffix trajectory $\tau_t = (s_t, a_t, \ldots, s_H)$, define its remaining cost as $C_t(\tau) = \sum_{k=t}^{H-1} c(s_k, a_k)$.

We define two conditioning strategies over supervision labels: *boundary-conditional supervision* uses trajectories within a narrow band $\mathcal{B}_t(\hat{C}_t, \epsilon)$, while *region-conditional supervision* includes all trajectories in the safe region $\mathcal{R}_t(\hat{C}_t)$, without modifying the model or training objective.

$$\mathcal{B}_t(\hat{C}_t, \epsilon) = \left\{ \tau_{t:} \mid C_t(\tau) \in [\hat{C}_t - \epsilon, \ \hat{C}_t + \epsilon] \right\}, \quad \mathcal{R}_t(\hat{C}_t) = \left\{ \tau_{t:} \mid C_t(\tau) < \hat{C}_t \right\}, \qquad (3)$$

where $\epsilon > 0$ is a small tolerance around the constraint threshold.

B2R achieves this through trajectory filtering and cost realignment. Filtering removes unsafe trajectories, ensuring feasibility, while realignment modifies costs to match the constraint token without changing state-action sequences. Additionally, we use rotary position embeddings (RoPE) to encode temporal structure, maintaining consistent conditioning across the safe region.

**Trajectory Filtering.**   To prevent unsafe trajectories from negatively impacting the policy, the B2R framework employs trajectory filtering as a preprocessing step. We define a *safe trajectory* as trajectory $\tau_{\text{safe}}$ satisfying $C(\tau) \le \kappa$, and define the corresponding safe dataset $\mathcal{D}_{\text{safe}}$ as:

$$\mathcal{D}_{\text{safe}} = \{ \tau \in \mathcal{D} \mid C(\tau) \le \kappa \}. \qquad (4)$$

This filtering step ensures that all training trajectories are compliant with the deployment constraint $\kappa$, enabling subsequent token conditioning to be consistent with test-time usage.

---

**Algorithm 1** Boundary-to-Region Framework

---

**Require:** Offline dataset $\mathcal{D}$, cost threshold $\kappa$, model parameters $\theta$

**Ensure:** Policy $\pi_\theta(a_t \mid \hat{R}_{0:t}, \hat{C}_{0:t}, s_{0:t}, a_{0:t-1})$

1: **1. Trajectory Filtering:**
2: Filter safe trajectories:
3: $\mathcal{D}_{\text{safe}} = \{ \tau \in \mathcal{D} \mid C(\tau) \le \kappa \}$
4: **2. CTG Realignment:**
5: **for** each $\tau \in \mathcal{D}_{\text{safe}}$ **do**
6:     Shift CTG: $\hat{C}'_t = \hat{C}_t + (\kappa - C(\tau))$
7: **end for**
8: **3. Tokenization with RoPE:**
9: For each timestep $t$, construct context:
10: $o_t = \left[ \hat{R}_{t-K:t}, \hat{C}'_{t-K:t}, s_{t-K:t}, a_{t-K:t-1} \right]$
11: Encode positions with RoPE
12: **4. Policy Training:**
13: Train Transformer to minimize:
14: $\mathcal{L}(\theta) = \mathbb{E}_{\tau \sim \mathcal{D}_{\text{safe}}} \left[ -\log \pi_\theta(a_t \mid o_t) \right]$
15: **return** $\pi_\theta$

---

**CTG Realignment.** As shown in Figure 3, conventional methods provide supervision only from trajectories whose cost lies near the constraint threshold (orange dots), leading to sparse and unstable learning signals. These methods discard large portions of feasible data simply because their cost does not exactly match the conditioning token.

In contrast, B2R leverages all safe trajectories by adjusting their costs to match the constraint, enabling consistent supervision under a fixed token while drawing on diverse behaviors originally scattered across the safe region (orange box). This process decouples the conditioning input from the supervision signal: $\hat{C}'_t = \kappa$. the model is consistently conditioned on a single boundary token, but learns to associate it with the diverse behaviors sourced from the entire safe region.

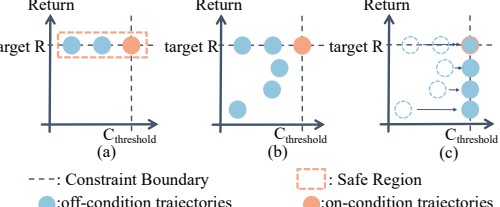

We adopt a realignment strategy, called **CTG-shift** to ensure consistency between each trajectory and the fixed initial CTG:

$$\hat{C}'_t = \hat{C}_t + (\kappa - C(\tau)). \quad (5)$$

Figure 3: Supervision strategy comparison. Conventional methods (a, b) rely on sparse, boundary-aligned trajectories. In contrast, B2R (c) realigns all safe trajectories to the constraint threshold (dashed line), transforming sparse boundary data into dense, region-wide supervision. Orange dots denote compliant trajectories; dashed arrows show the realignment.

Here the shifted CTG sequence $\{\hat{C}'_t\}$ is constructed by adding a constant offset $\kappa - C(\tau)$ to each step, such that $\hat{C}'_0 = \kappa$, and the temporal profile of the original CTG is preserved. This adjustment ensures that the entire sequence remains strictly decreasing, and that each supervision token is semantically aligned with the deployment-time constraint. We compare four CTG realignment strategies: **Shift** (uniform offset), **Avg** (even redistribution), **Rand** (random redistribution), and **Scale** (multiplicative normalization). Detailed discussion are provided in Appendix B.1.

**Model and Inference.** The policy model in B2R adopts a transformer-based autoregressive structure, augmented with RoPE, whose relative positional encoding is better suited for capturing the step-by-step cost dynamics introduced by our CTG realignment strategy. As shown in Figure 4, these tokens are processed by a Transformer encoder, which outputs the mean $\mu_t$ of a Gaussian action distribution $\pi_\theta(a_t \mid \cdot) = \mathcal{N}(\mu_t, \sigma_t^2)$. The model is trained via behavior cloning on CTG-aligned trajectories using a fixed initial CTG $\hat{C}'_0 = \kappa$, and is queried autoregressively at inference with consistent input, ensuring alignment with the deployment-time safety budget. The model minimizes negative log-likelihood of action:

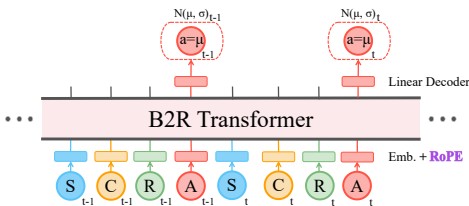

Figure 4: Architecture of the Transformer model in the B2R framework. The model takes tokenized inputs consisting of states, actions, RTG, and CTG, and augments them with RoPE for improved temporal modeling.

$$\mathcal{L}_{\text{BC}}(\theta) = \mathbb{E}_{\tau^{(n)} \sim \mathcal{D}_{\text{safe}}} \left[ -\log \pi_\theta \left( a_t^{(n)} \mid \hat{R}_{t-K:t}^{(n)}, \hat{C}'_{t-K:t}^{(n)}, s_{t-K:t}^{(n)}, a_{t-K:t-1}^{(n)} \right) \right] \quad (6)$$

### 4.3 Theoretical Analysis

While B2R is conceptually simple, it fundamentally reshapes how RvS agents generalize under constraints. To explore how B2R balances reward maximization with cost constraint satisfaction, we analyze its theoretical foundations, focusing on the relationship between the learned policy's return and its adherence to the cost constraint. We first focus on safety adherence, analyzing how B2R ensures cost constraint satisfaction while maintaining a safety margin. This analysis is built upon Assumptions 1 and 2, leading to the results presented in Theorem 1.

**Assumption 1** (Safe-Aligned Data) After CTG realignment every training trajectory obeys

$$\hat{C}'_0 = \kappa, \quad \hat{C}'_{t+1} = \hat{C}'_t - c_t, \quad 0 \le c_t \le C_{\max}, \quad (7)$$

Essentially, this ensures the agent never observes "unsafe" cost patterns during training—it learns from data where budget consumption evolves as if the agent were truly constrained. Additionally, we

assume that the dataset used during training remains static, ensuring consistent training data over the course of the learning process.

**Assumption 2** (Prediction-Error Bound). At deployment, the agent's implicit cost predictions $\hat{c}_t$ (conditioned on trajectory history $\tau_n^t$) must be sufficiently accurate:

$$\mathbb{E}\big[|c_t - \hat{c}_t|\big] \; \leq \; \sigma, \quad 0 < \sigma H < \delta, \quad \sum_{t=0}^{H-1} \hat{c}_t \; \leq \; \kappa - \tfrac{\delta}{2}. \tag{8}$$

Here $\sigma$ bounds the per-step prediction error, while $\delta$ ensures the planned cost stays safely below $\kappa$ (accounting for potential error accumulation). The requirement $\sigma H < \delta$ prevents error compounding from overwhelming the safety margin. In practice, this assumption can be violated if the agent encounters highly out-of-distribution states (leading to a large single-step error that breaches the $\sigma$ bound) or if its planned trajectory is overly optimistic, leaving an insufficient safety margin $\sigma$ to account for cumulative errors. This assumption is based on the premise that the model has sufficient expressive capacity to accurately predict the cost and can learn to replicate the policy.

Under Assumptions 1-2, B2R provides probabilistic and expected-cost safety guarantees:

**Theorem 1**(Safety Guarantees of B2R). Let $\tau$ be generated with initial CTG $\hat{C}_0 = \kappa$. Then:

$$\Pr\big[C^{\text{B2R}}(\tau) \leq \kappa\big] \; \geq \; 1 - \exp\left(-\tfrac{(\delta - \sigma H)^2}{2H C_{\max}^2}\right), \quad \mathbb{E}\big[C^{\text{B2R}}(\tau)\big] \; \leq \; \kappa - (\delta - \sigma H). \tag{9}$$

Here, $\sigma$ denotes an upper bound on the per-step cost prediction error during deployment. The proof is in Appendix A.1. Equation 9 provides a dual guarantee of high–probability and expected safety at the trajectory level for B2R: the **probabilistic guarantee** shows that larger safety margins $\delta$ make constraint violations exponentially rare. The **cost expectation** bound ensures the average trajectory stays $\delta - \sigma H$ below the budget, even under worst-case error.

We now analyze B2R's performance in terms of maximum cumulative reward (Equation 1). Unlike safety, which minimizes cost, our focus here is on whether enforcing safety degrades reward. Under the following assumption, B2R can preserve or even improve reward:

**Assumption 3 (Optimal-Coverage).** The dataset contains a safe trajectory $\tau^\star$ such that

$$C(\tau^\star) < \kappa \quad \text{and} \quad R(\tau^\star) = \max_{\tau \in D_{\text{safe}}} R(\tau) \tag{10}$$

This assumption ensures that the return-optimal trajectory is within the feasible region, so B2R's filtering and realignment do not exclude it from training.

**Theorem 2 (Performance Superiority of B2R over Boundary Supervision).** Under Assumptions 1 and 3, B2R guarantees:

$$R_{\max}^{\text{B2R}}(\kappa) \geq R_{\max}^{\text{boundary}}(\kappa) \tag{11}$$

That is, B2R achieves reward no worse than boundary-conditional supervision while satisfying $C(\tau) \leq \kappa$. The proof is provided in Appendix A.2.

# 5 Experiments

We evaluate the B2R framework on two fronts: (1) its ability to balance reward maximization and safety compliance under cost constraints, and (2) the individual impact of its key components, particularly cost realignment. We also examine B2R's effectiveness in selecting feasible actions compared to baseline methods.

**Environments.** Experiments are conducted on the DSRL benchmark [25], which includes 38 sequential decision-making tasks of varying difficulty. This suite provides a diverse and realistic testbed for safety-critical offline RL. Full environment details are in Appendix C.2.

**Metrics.** We report both reward and cost, aiming to maximize the former while keeping the latter below the constraint threshold. Each method is evaluated under three cost limits and three random seeds per task. Normalization details and metric definitions are provided in Appendix C.3.

**Baselines.** We compare our approach with several state-of-the-art algorithms, including BC-All, BC-Safe, CDT [26], BCQ-Lag [8], CPQ [34], COptiDICE [19], and TraC [10]. Some experimental results are obtained from the DSRL implementation, while others are sourced from the official codebase. See Appendix C.4 for details.

Table 1: Normalized reward and cost results. ↑ indicates that higher values are better, while ↓ indicates that lower values are preferable. Metrics are averaged over 3 cost thresholds, 20 evaluation episodes, and 3 random seeds. **Bold** marks safe agents with normalized costs below 1, while **Blue** highlights safe agents achieving the highest reward.

| Task | BC-All | | BC-Safe | | CDT | | BCQ-Lag | | CPQ | | COptiDICE | | TraC | | B2R(ours) | |
|---|---|---|---|---|---|---|---|---|---|---|---|---|---|---|---|---|
| | reward↑ | cost↓ | reward↑ | cost↓ | reward↑ | cost↓ | reward↑ | cost↓ | reward↑ | cost↓ | reward↑ | cost↓ | reward↑ | cost↓ | reward↑ | cost↓ |
| PointButton1 | 0.1 | 1.05 | **0.06** | **0.52** | 0.53 | 1.68 | 0.24 | 1.73 | 0.69 | 3.2 | 0.13 | 1.35 | **0.17** | **0.91** | **0.19** | **0.96** |
| PointButton2 | 0.27 | 2.02 | 0.16 | 1.1 | 0.46 | 1.57 | 0.4 | 2.66 | 0.58 | 4.3 | 0.15 | 1.51 | **0.16** | **0.91** | **0.16** | **0.91** |
| PointCircle1 | 0.79 | 3.98 | **0.41** | **0.16** | **0.59** | **0.69** | 0.54 | 2.38 | **0.43** | **0.75** | 0.86 | 5.51 | **0.50** | **0.07** | 0.54 | 0.31 |
| PointCircle2 | 0.66 | 4.17 | **0.48** | **0.99** | 0.64 | 1.05 | 0.66 | 2.6 | 0.24 | 3.58 | 0.85 | 8.61 | **0.61** | **0.86** | **0.61** | **0.80** |
| PointGoal1 | **0.65** | **0.95** | **0.43** | **0.54** | 0.69 | 1.12 | **0.71** | **0.98** | 0.57 | 0.35 | 0.49 | 1.66 | **0.44** | **0.36** | **0.58** | **0.71** |
| PointGoal2 | 0.54 | 1.97 | **0.29** | **0.78** | 0.59 | 1.34 | 0.67 | 3.18 | 0.4 | 1.31 | 0.38 | 1.92 | **0.31** | **0.59** | **0.34** | **0.68** |
| PointPush1 | **0.19** | **0.61** | **0.24** | **0.43** | **0.24** | **0.48** | **0.33** | **0.86** | 0.2 | 0.83 | **0.13** | **0.83** | **0.15** | **0.42** | **0.22** | **0.67** |
| PointPush2 | **0.18** | **0.91** | **0.11** | **0.8** | **0.21** | **0.65** | **0.23** | **0.99** | 0.11 | 1.04 | 0.02 | 1.18 | **0.15** | **0.8** | **0.16** | **0.76** |
| CarButton1 | 0.03 | 1.38 | **0.07** | **0.85** | 0.21 | 1.6 | 0.04 | 1.63 | 0.42 | 9.66 | -0.08 | 1.68 | **-0.03** | **0.59** | **0.04** | **0.65** |
| CarButton2 | -0.13 | 1.24 | **-0.01** | **0.63** | 0.13 | 1.58 | 0.06 | 2.13 | 0.37 | 12.51 | -0.07 | 1.59 | **-0.08** | **0.62** | **-0.01** | **0.62** |
| CarCircle1 | 0.72 | 4.39 | 0.37 | 1.38 | 0.6 | 1.73 | 0.73 | 5.25 | 0.02 | 2.29 | 0.7 | 5.72 | 0.52 | 1.85 | 0.51 | 2.14 |
| CarCircle2 | 0.76 | 6.44 | 0.54 | 3.38 | 0.66 | 2.53 | 0.72 | 6.58 | 0.44 | 2.69 | 0.77 | 7.99 | 0.59 | 2.33 | 0.42 | 1.90 |
| CarGoal1 | **0.39** | **0.33** | **0.24** | **0.28** | 0.66 | 1.21 | **0.47** | **0.78** | 0.79 | 1.42 | **0.35** | **0.54** | **0.38** | **0.39** | **0.52** | **0.72** |
| CarGoal2 | 0.23 | 1.05 | **0.14** | **0.51** | 0.48 | 1.25 | 0.3 | 1.44 | 0.65 | 3.75 | **0.25** | **0.91** | **0.19** | **0.52** | **0.22** | **0.66** |
| CarPush1 | **0.22** | **0.36** | **0.14** | **0.33** | **0.31** | **0.4** | **0.23** | **0.43** | -0.03 | 0.95 | **0.23** | **0.5** | **0.19** | **0.18** | **0.28** | **0.56** |
| CarPush2 | **0.14** | **0.9** | **0.05** | **0.45** | 0.19 | 1.3 | 0.15 | 1.38 | 0.24 | 4.25 | 0.09 | 1.07 | **0.08** | **0.54** | **0.11** | **0.79** |
| SwimmerVelocity | 0.49 | 4.72 | 0.51 | 1.07 | **0.66** | **0.96** | 0.48 | 6.58 | 0.13 | 2.66 | 0.63 | 7.58 | 0.55 | 3.21 | **0.49** | **0.46** |
| HopperVelocity | 0.65 | 6.39 | **0.36** | **0.67** | **0.63** | **0.61** | 0.78 | 5.02 | 0.14 | 2.11 | 0.13 | 1.51 | **0.57** | **0.98** | **0.63** | **0.53** |
| HalfCheetahVelocity | 0.97 | 13.1 | **0.88** | **0.54** | **1.0** | **0.01** | 1.05 | 18.21 | **0.29** | **0.74** | **0.65** | **0.0** | 0.96 | 2.5 | **0.95** | **0.00** |
| Walker2dVelocity | 0.79 | 3.88 | **0.79** | **0.04** | **0.78** | **0.06** | **0.79** | **0.17** | **0.04** | **0.21** | **0.12** | **0.74** | **0.64** | **0.06** | **0.79** | **0.01** |
| AntVelocity | 0.98 | 3.72 | **0.98** | **0.29** | **0.98** | **0.39** | 1.02 | 4.15 | -1.01 | 0.0 | 1.0 | 3.28 | **0.97** | **0.15** | **0.99** | **0.42** |
| **SafetyGym Average** | 0.46 | 3.03 | **0.34** | **0.75** | 0.54 | 1.06 | 0.5 | 3.29 | 0.27 | 2.79 | 0.37 | 2.65 | **0.40** | **0.92** | **0.42** | **0.73** |
| BallRun | 0.6 | 5.08 | 0.27 | 1.46 | 0.39 | 1.16 | 0.76 | 3.91 | 0.22 | 1.27 | 0.59 | 3.52 | **0.27** | **0.47** | **0.31** | **0.23** |
| CarRun | 0.97 | 0.33 | **0.94** | **0.22** | **0.99** | **0.65** | **0.94** | **0.15** | 0.95 | 1.79 | **0.87** | **0.0** | **0.97** | **0.03** | **0.96** | **0.08** |
| DroneRun | 0.24 | 2.13 | **0.28** | **0.74** | **0.63** | **0.79** | 0.42 | 2.47 | 0.33 | 3.52 | 0.67 | 4.15 | **0.55** | **0.01** | **0.56** | **0.03** |
| AntRun | 0.72 | 2.93 | 0.65 | 1.09 | **0.72** | **0.91** | 0.76 | 5.11 | **0.03** | **0.02** | **0.61** | **0.94** | **0.67** | **0.63** | **0.72** | **0.69** |
| BallCircle | 0.74 | 4.71 | **0.52** | **0.65** | 0.77 | 1.07 | 0.69 | 2.36 | **0.64** | **0.76** | 0.7 | 2.61 | **0.68** | **0.59** | **0.67** | **0.59** |
| CarCircle | 0.58 | 3.74 | **0.5** | **0.84** | **0.75** | **0.95** | 0.63 | 1.89 | **0.71** | **0.33** | 0.49 | 3.14 | **0.64** | **0.76** | **0.71** | **0.68** |
| DroneCircle | 0.72 | 3.03 | **0.56** | **0.57** | **0.63** | **0.98** | 0.8 | 3.07 | -0.22 | 1.28 | 0.26 | 1.02 | **0.6** | **0.67** | **0.57** | **0.34** |
| AntCircle | 0.58 | 4.9 | **0.4** | **0.96** | 0.54 | 1.78 | 0.58 | 2.87 | **0.0** | **0.0** | 0.17 | 5.04 | **0.47** | **0.98** | 0.45 | 1.36 |
| **BulletGym Average** | 0.64 | 3.36 | **0.52** | **0.82** | 0.68 | 1.04 | 0.74 | 3.11 | 0.33 | 1.12 | 0.55 | 2.55 | **0.61** | **0.52** | **0.62** | **0.50** |
| easysparse | 0.17 | 1.54 | **0.11** | **0.21** | **0.17** | **0.23** | 0.78 | 5.01 | **-0.06** | **0.07** | 0.96 | 5.44 | **0.35** | **0.05** | **0.77** | **0.70** |
| easymean | 0.43 | 2.82 | **0.04** | **0.29** | **0.45** | **0.54** | 0.71 | 3.44 | **-0.07** | **0.07** | 0.66 | 3.97 | **0.29** | **0.05** | **0.77** | **0.69** |
| easydense | 0.27 | 1.94 | **0.11** | **0.14** | **0.32** | **0.62** | 0.26 | 0.47 | **-0.06** | **0.03** | 0.5 | 2.54 | **0.24** | **0.06** | **0.76** | **0.69** |
| mediumsparse | 0.83 | 3.34 | **0.33** | **0.3** | 0.87 | 1.1 | 0.44 | 1.16 | **-0.08** | **0.07** | 0.71 | 2.49 | **0.34** | **0.06** | **0.92** | **0.58** |
| mediummean | 0.77 | 2.53 | **0.31** | **0.21** | **0.45** | **0.75** | 0.78 | 1.53 | **-0.08** | **0.05** | 0.76 | 2.05 | **0.32** | **0.06** | **0.88** | **0.63** |
| mediumdense | 0.45 | 1.47 | **0.24** | **0.17** | 0.88 | 2.41 | 0.58 | 1.89 | **-0.07** | **0.07** | 0.69 | 2.24 | **0.33** | **0.06** | **0.92** | **0.70** |
| hardsparse | 0.42 | 1.8 | 0.17 | 3.25 | **0.25** | **0.41** | 0.5 | 1.02 | **-0.05** | **0.06** | 0.37 | 2.05 | **0.41** | **0.03** | **0.51** | **0.48** |
| hardmean | 0.2 | 1.77 | **0.13** | **0.4** | **0.33** | **0.97** | 0.47 | 2.56 | **-0.05** | **0.06** | 0.32 | 2.47 | **0.44** | **0.05** | **0.50** | **0.58** |
| harddense | 0.2 | 1.33 | **0.15** | **0.22** | **0.08** | **0.21** | 0.35 | 1.4 | **-0.04** | **0.08** | 0.24 | 1.68 | **0.39** | **0.06** | **0.48** | **0.58** |
| **MetaDrive Average** | 0.42 | 2.06 | **0.18** | **0.58** | 0.42 | 0.8 | 0.54 | 2.05 | **-0.06** | **0.06** | 0.58 | 2.77 | **0.35** | **0.05** | **0.72** | **0.62** |

## 5.1 Results on DSRL Benchmark

Table 1 shows that B2R effectively balances reward and cost, satisfying safety constraints in 35 out of 38 tasks and achieving the highest rewards in 20. It also ranks first in average score across Safety-Gymnasium, Bullet Safety-Gym, and MetaDrive. B2R's robustness is notable, though it struggles in a few exceptionally challenging environments like CarCircle1/2 and AntCircle, where most baselines also fail. These failures are attributable to the datasets: high-reward trajectories are coupled with high-cost actions, leaving the offline data with insufficient examples of the precise, long-horizon control required to navigate the narrow safe action space near unforgiving boundaries.

Compared to Transformer-based methods like CDT, B2R achieves lower cost in most environments while maintaining competitive reward. CDT's sensitivity to tight safety constraints stems from its reliance on sparse boundary-conditional supervision. This fundamental limitation cannot be reliably solved by simply training with a stricter, buffered cost threshold—an approach that our experiments show leads to an unstable safety-performance trade-off. In contrast, B2R's cost realignment enables region-wide supervision, drawing from a denser and more diverse set of safe trajectories to ensure robust and consistent constraint adherence. A detailed analysis of the buffered baseline comparison is available in Appendix B.3.

B2R also demonstrates superior performance over methods like BC-Safe and TraC. While all approaches begin by filtering for safe trajectories, the fundamental difference lies in how they utilize the resulting safe data. BC-Safe learns from a sparse signal, while TraC relies on a coarse, binary classification of trajectories into "desirable" and "undesirable" sets. In contrast, B2R's core innovation is to realign all safe trajectories, which transforms the learning problem into a fine-grained temporal regression task. This creates a dense supervision signal that allows the Decision Transformer framework to better learn the causal dynamics between actions and future costs, enabling more effective adaptation in diverse safety-critical settings.

Figure 5 presents B2R's average reward and cost at each constraint level (L1–L3), grouped by benchmark suite. In MetaDrive, we observe a non-monotonic cost trend: cost increases from L1 to

L2 before dropping at L3. This is likely because L1 filtering excludes many high-cost trajectories, biasing the dataset toward safer behavior at the lowest constraint. Overall, B2R maintains stable or improving rewards and shows decreasing cost under tighter constraints in most cases, demonstrating its robustness across varying safety levels.

Furthermore, to stress-test B2R's robustness under particularly stringent cost limits, we conducted a direct comparison against FISOR using its original, stricter evaluation protocol. The results, detailed in Appendix B.6, show that B2R achieves a superior safety-performance balance even in these data-scarce scenarios.

Figure 5: Average B2R performance on Bullet-SafetyGym (BS), SafetyGymnasium (SG), and MetaDrive (MD) across three constraint levels (L1-L3). Tighter constraints generally decrease cost while rewards remain stable or improve. The non-monotonic cost trend in MD is likely an artifact of L1 filtering bias (see Appendix C.3 for thresholds).

## 5.2 Ablation of B2R Components

We ablate three core components of B2R—CTG realignment, RoPE, and trajectory filtering—each removed independently while keeping the rest of the system fixed (Figure 6). We evaluate their impact on reward and safety across tasks from three environment groups covering diverse dynamics and constraint structures.

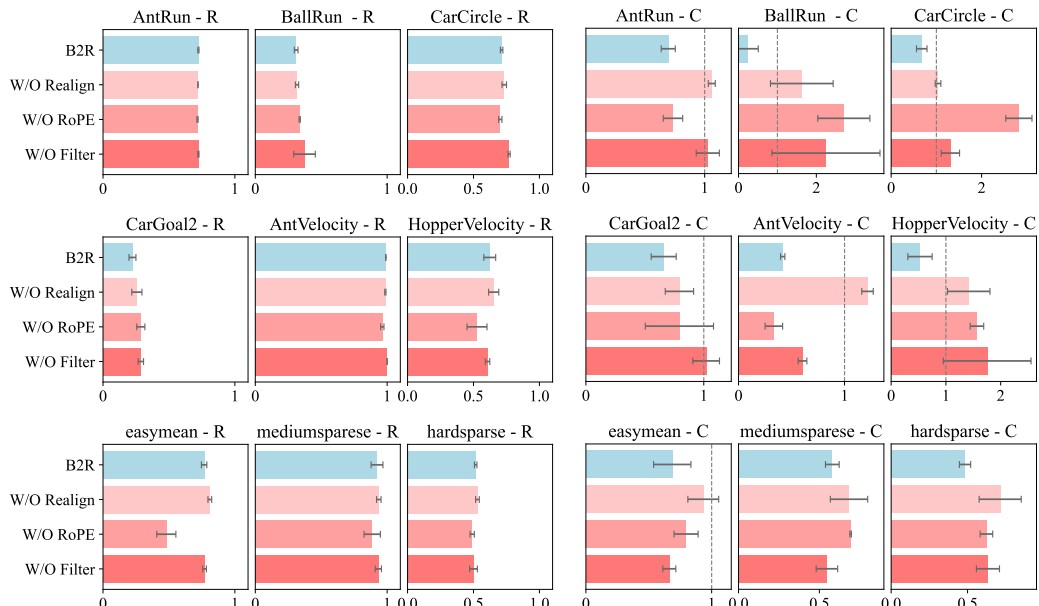

Figure 6: Ablation study of B2R across 9 tasks from the DSRL benchmark. We assess the effect of three components: cost-to-go realignment (W/O Realign), trajectory filtering (W/O Filter), and our choice of positional embeddings. For the latter, we compare our default RoPE against standard absolute positional embeddings, which is labeled as 'W/O RoPE' in the plots. The full B2R model achieves favorable reward and cost performance compared to ablated variants, highlighting the contribution of each module.

**CTG Realignment.** Removing CTG realignment results in higher cumulative cost across nearly all environments, with frequent violations of the safety threshold (e.g., `BallRun`, `CarCircle`, `AntVelocity`). Reward remains identical to the full model, aligning with our theoretical expectation: CTG realignment alters only the cost signal while leaving the reward structure untouched.

**Positional Embeddings.** The results show that replacing RoPE with standard absolute positional embeddings (APE, labeled as 'W/O RoPE' in Figure 6) leads to significant performance drops in tasks such as `CarCircle`, `HopperVelocity`, and `easymean`, with both higher costs and lower rewards.

This empirically confirms that RoPE's ability to capture relative temporal dependencies is more effective for modeling the fine-grained, step-by-step cost dynamics inherent in our framework, a task for which APE is less suited.

**Trajectory Filtering.** Removing trajectory filtering causes a marked increase in cost and more unstable performance across seeds, particularly in tasks such as `CarCircle`, `HopperVelocity`, and `hardsparse`. While reward may slightly improve in some settings due to exposure to higher-reward but unsafe data, the model frequently violates constraints. This trade-off highlights filtering's role in enforcing feasible supervision and stabilizing constraint satisfaction.

## 5.3 Cost-Aware Action Selection under Aligned Supervision

To examine how B2R improves constraint satisfaction, we compare its action distribution with a baseline in the `easysparse` environment using t-SNE (Figure 7). Actions are marked unsafe if their one-step cost exceeds the per-step budget. The baseline produces a notable number of unsafe actions, scattered across distinct state–action clusters. This suggests that boundary-aligned supervision provides limited coverage, causing the model to associate cost only with a narrow set of behaviors and generalize poorly to high-risk scenarios. In contrast, B2R reduces unsafe actions and yields more concentrated, constraint-compliant responses. Its action clusters remain largely within the safe region, indicating better generalization to the intended constraint threshold.

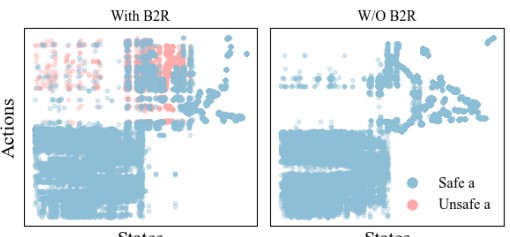

Figure 7: t-SNE visualization of actions from the baseline (right) and B2R (left) in the `easysparse` environment (threshold = 10). Actions are labeled as safe (blue) if their one-step cost is below the threshold divided by episode length, and unsafe (red) otherwise. B2R selects fewer unsafe actions than the baseline, reflecting its cost-aligned training under a unified constraint.

## 6 Conclusion and Future Work

In this work, we proposed Boundary-to-Region, a simple yet effective framework for offline safe reinforcement learning. By addressing the reward–cost asymmetry, B2R realigns the cost-to-go tokens of all trajectories to reflect the desired safety threshold and filters out infeasible behaviors, ensuring the model learns safe, cost-compliant policies. Extensive experiments on benchmark tasks validate B2R's effectiveness, achieving leading performance in 20 environments while maintaining safety compliance in 35 out of 38 cases. These results demonstrate B2R's ability to optimize rewards under strict constraints while ensuring safety.

A key consideration for B2R is its reliance on the availability of high-quality safe trajectories, a common challenge in offline safe RL. In environments where safe trajectories are exceedingly rare, B2R's performance may degrade due to the reduced size of the training set. To investigate this, we conducted few-shot experiments (detailed in Appendix B.4), which show that B2R exhibits a graceful degradation profile and remains data-efficient due to its region-wide supervision. Nevertheless, for extreme cases of data scarcity, future work could explore integrating B2R with data augmentation or generative modeling techniques to synthesize diverse, safe trajectories.

While the non-adaptive CTG realignment strategy used in this work effectively validates our core paradigm, a promising future direction is to explore learned realignment strategies. An adaptive mechanism could, for example, allocate the safety margin non-uniformly to better handle high-risk states, offering more fine-grained control. We view this as a valuable extension of the B2R framework.

Additionally, while this work focuses on a single safety threshold for clarity and practical relevance, our framework is extensible to multi-target scenarios. We present a proof-of-concept in Appendix B.5, demonstrating that a single B2R agent can be trained to adhere to various constraint levels without retraining while maintaining comparable performance.

## Acknowledgements

I wish to thank the anonymous reviewers for their constructive comments, which have greatly improved the quality of this paper. Also, many thanks to YM. An, my life partner. She gently eased all the anxiety and pressure born from this work. Her presence is what gave meaning to the entire endeavor.

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

# Appendix

## Table of Contents

## A  Proofs

### A.1  Proof of Theorem 1

We begin by establishing notation and core relationships. Let $e_t := c_t - \hat{c}_t$ denote the instantaneous cost estimation error, and define the cumulative estimation error $D_t := \sum_{i=0}^{t-1} e_i$. The filtration $\mathcal{F}_t$ captures all historical information up to time $t$, specifically $\mathcal{F}_t := \sigma(s_0, a_0, \hat{C}_0, \ldots, s_t, a_t, \hat{C}_t)$.

**Step 1: Cost Accounting via Telescoping Sums** The CTG mechanism maintains the budget estimate $\hat{C}_t$ through the recursion:

$$\hat{C}_{t+1} = \hat{C}_t - c_t, \quad \hat{C}_0 = \kappa \tag{12}$$

Telescoping this relationship over $H$ steps reveals:

$$\sum_{t=0}^{H-1} c_t = \hat{C}_0 - \hat{C}_H = \kappa - \hat{C}_H \tag{13}$$

Consequently, the total realized cost $C(\tau) = \kappa - \hat{C}_H$, with budget violation $C(\tau) > \kappa$ equivalent to $\hat{C}_H < 0$.

**Step 2: Martingale Construction for Error Propagation** We analyze the cumulative error $D_t$ through the martingale difference sequence:

$$M_t := D_t - \mathbb{E}[D_t | \mathcal{F}_{t-1}], \quad M_0 = 0 \tag{14}$$

The martingale property $\mathbb{E}[M_t | \mathcal{F}_{t-1}] = M_{t-1}$ follows immediately from the tower property of conditional expectation.

**Step 3: Bounding Martingale Increments** The difference sequence satisfies:

$$|M_{t+1} - M_t| \le |e_t| + \mathbb{E}[|e_t| | \mathcal{F}_t] \le 2C_{\max} =: \tilde{C}_{\max} \tag{15}$$

where the first inequality uses the triangle inequality and the second follows from our bounded cost assumption $|e_t| \le C_{\max}$ combined with Jensen's inequality.

**Step 4: Concentration via Azuma-Hoeffding Inequality** Applying the Azuma-Hoeffding inequality to the martingale $\{M_t\}$ yields:

$$\Pr[M_H \geq \eta] \leq \exp\left(-\frac{\eta^2}{2H\tilde{C}_{\max}^2}\right) \tag{16}$$

Setting $\eta = \varepsilon - \sigma H$ (positive by assumption $\sigma H < \varepsilon$) gives the probability bound:

$$\delta = \exp\left(-\frac{(\varepsilon - \sigma H)^2}{2HC_{\max}^2}\right) \tag{17}$$

**Step 5: Error Control Under Good Event** Define the favorable event $\mathcal{E} := \{M_H < \varepsilon - \sigma H\}$. On this event:

$$D_H = M_H + \mathbb{E}[D_H | \mathcal{F}_{H-1}] \tag{18}$$

$$< (\varepsilon - \sigma H) + \sum_{t=0}^{H-1} \mathbb{E}[e_t | \mathcal{F}_{H-1}] \tag{19}$$

$$\leq \varepsilon \tag{20}$$

where 20 follows from our behavioral cloning assumption $\mathbb{E}[|e_t|| \mathcal{F}_{H-1}] \leq \sigma$.

**Step 6: Conservative Budget Planning** By the training data constraint $C_{\text{data}}(\tau) \leq \kappa - \varepsilon$ and $\sigma$-accurate cost estimation:

$$\sum_{t=0}^{H-1} \hat{c}_t \leq \kappa - \frac{\varepsilon}{2} \tag{21}$$

**Step 7: Probabilistic Safety Guarantee** Combining results on event $\mathcal{E}$:

$$C(\tau) = \sum_{t=0}^{H-1} \hat{c}_t + D_H \tag{22}$$

$$\leq \left(\kappa - \frac{\varepsilon}{2}\right) + \varepsilon = \kappa + \frac{\varepsilon}{2} \tag{23}$$

Thus the probability of budget violation satisfies:

$$\Pr[C(\tau) > \kappa] \leq \Pr[\mathcal{E}^{\complement}] \leq \exp\left(-\frac{(\varepsilon - \sigma H)^2}{2HC_{\max}^2}\right) \tag{24}$$

**Step 8: Expected Cost Analysis** Finally, the unbiasedness of cost estimates $\mathbb{E}[e_t] = 0$ implies:

$$\mathbb{E}[C(\tau)] = \sum_{t=0}^{H-1} \mathbb{E}[\hat{c}_t] + \mathbb{E}[D_H] \tag{25}$$

$$\leq \kappa - (\varepsilon - \sigma H) \tag{26}$$

completing the proof.

### A.2 Proof of Theorem 2

**Step 1: Supervision Set Inclusion (Based on Definition 1)** By Definition 1, boundary-conditional supervision uses trajectories in:

$$\mathcal{B}_t(\hat{C}_t, \epsilon) = \left\{\tau_{t:} \mid C_t(\tau) \in [\hat{C}_t - \epsilon, \hat{C}_t + \epsilon]\right\} \tag{27}$$

while region-conditional supervision (B2R) uses:

$$\mathcal{R}_t(\hat{C}_t) = \left\{\tau_{t:} \mid C_t(\tau) < \hat{C}_t\right\} \tag{28}$$

After filtering, for the entire trajectory ($t = 0$), this implies:

$$\mathcal{T}_{\text{boundary}} = \{\tau \mid C(\tau) \in [\kappa - \epsilon,\ \kappa]\} \subseteq \mathcal{T}_{\text{region}} = \{\tau \mid C(\tau) \leq \kappa\} \tag{29}$$

*Implication:* B2R retains all trajectories available to boundary supervision.

**Step 2: Maximization Scope** Let

$$\tau^*_{\text{boundary}} = \arg \max_{\tau \in \mathcal{T}_{\text{boundary}}} R(\tau) \tag{30}$$

By Assumption 3, there exists $\tau^\star \in \mathcal{T}_{\text{region}}$ with:

$$C(\tau^\star) < \kappa \quad \text{and} \quad R(\tau^\star) = \max_{\tau \in \mathcal{D}_{\text{safe}}} R(\tau) \tag{31}$$

Two cases arise:

1. If $\tau^\star \in \mathcal{T}_{\text{boundary}}$, then
$$R_{\max}^{\text{B2R}}(\kappa) = R(\tau^\star) = R_{\max}^{\text{boundary}}(\kappa) \tag{32}$$

2. If $\tau^\star \notin \mathcal{T}_{\text{boundary}}$, then
$$R_{\max}^{\text{B2R}}(\kappa) = R(\tau^\star) > R_{\max}^{\text{boundary}}(\kappa) \tag{33}$$

**Step 3: Strict Improvement via Safe Region** Case 2 leverages Assumption 3: the existence of $\tau^\star \in \mathcal{T}_{\text{region}} \setminus \mathcal{T}_{\text{boundary}}$ ensures:

$$R_{\max}^{\text{B2R}}(\kappa) = R(\tau^\star) > R_{\max}^{\text{boundary}}(\kappa) \tag{34}$$

This aligns with B2R's ability to exploit strictly safer, higher-reward trajectories excluded by boundary supervision.

**Step 4: Guaranteed Lower Bound** Even if $\tau^\star \in \mathcal{T}_{\text{boundary}}$, set inclusion ensures:

$$R_{\max}^{\text{B2R}}(\kappa) \geq R_{\max}^{\text{boundary}}(\kappa) \tag{35}$$

B2R cannot perform worse as it includes all boundary-supervised trajectories.

***Remark.*** To simplify the analysis, we do not consider trajectory stitching. As a result, B2R conservatively learns only from full trajectories that satisfy the safety threshold $C(\tau) \leq \kappa$, ensuring that supervision remains within the feasible region throughout training.

# B  Supplementary experiments

## B.1  CTG Realignment Strategies

While B2R aligns CTG tokens to a fixed deployment-time budget, the choice of realignment strategy remains flexible. Our main results use a simple **Shift** method that applies a uniform offset to each trajectory's CTG sequence. However, alternative designs—based on assumptions such as uniformity, proportionality, or stochasticity—may influence reward optimization and constraint satisfaction differently.

To investigate this, we evaluate four representative strategies, each reflecting a distinct inductive bias in how cost should be redistributed. All methods enforce $\hat{C}'_0 = \kappa$, but vary in how they shape the remaining cost sequence. Below, we summarize their design principles:

**Shift (default).** The Shift strategy adds a constant offset $\Delta = \kappa - C(\tau)$ to all cost-to-go tokens along the trajectory:

$$\hat{C}'_t = \hat{C}_t + \Delta. \tag{36}$$

This preserves the original temporal profile of cost decay, maintaining consistency with the behavior's natural cost progression. It assumes that preserving cost shape is helpful for learning, and only alignment at the start token is necessary.

**Avg.** The Avg strategy evenly distributes the total offset $\Delta = \kappa - C(\tau)$ across all steps of the trajectory, modifying per-step costs as:

$$c'_t = c_t + \frac{\Delta}{H}, \quad \text{for all } t = 0, \dots, H - 1, \tag{37}$$

where $H$ is the trajectory length. The updated cost-to-go sequence is then recomputed as:

$$\hat{C}'_t = \sum_{k=t}^{H-1} c'_k. \tag{38}$$

This approach enforces uniform per-step adjustment, flattening cost variation across time. It assumes that smoothing cost signals may stabilize learning, especially in environments with noisy or sparse cost feedback.

**Rand.** Rand randomly reallocates the excess budget across eligible timesteps. In discrete environments (e.g., SafetyGym), this involves flipping randomly chosen $c_t = 0$ steps to $c'_t = 1$ until $\Delta$ is exhausted. In continuous environments (e.g., MetaDrive), we sample $c_t < \kappa/H$ and update:

$$c'_t = \frac{\kappa}{H}, \quad \Delta \leftarrow \Delta - \left( \frac{\kappa}{H} - c_t \right). \tag{39}$$

The CTG is then recomputed as:

$$\hat{C}'_t = \sum_{k=t}^{H} c'_k. \tag{40}$$

This method introduces stochasticity, testing the model's robustness to non-smooth cost supervision.

**Scale.** The Scale strategy rescales the entire CTG sequence by a multiplicative factor $\alpha = \kappa/C(\tau)$:

$$\hat{C}'_t = \alpha \cdot \hat{C}_t. \tag{41}$$

This preserves the relative shape of the cost curve while adjusting its magnitude. It assumes that cost proportionality, rather than absolute values, is the key for generalization under a fixed constraint.

Each strategy ensures $\hat{C}'_0 = \kappa$, but their internal structure introduces distinct biases that may affect policy behavior. The following sections compare their empirical effects.

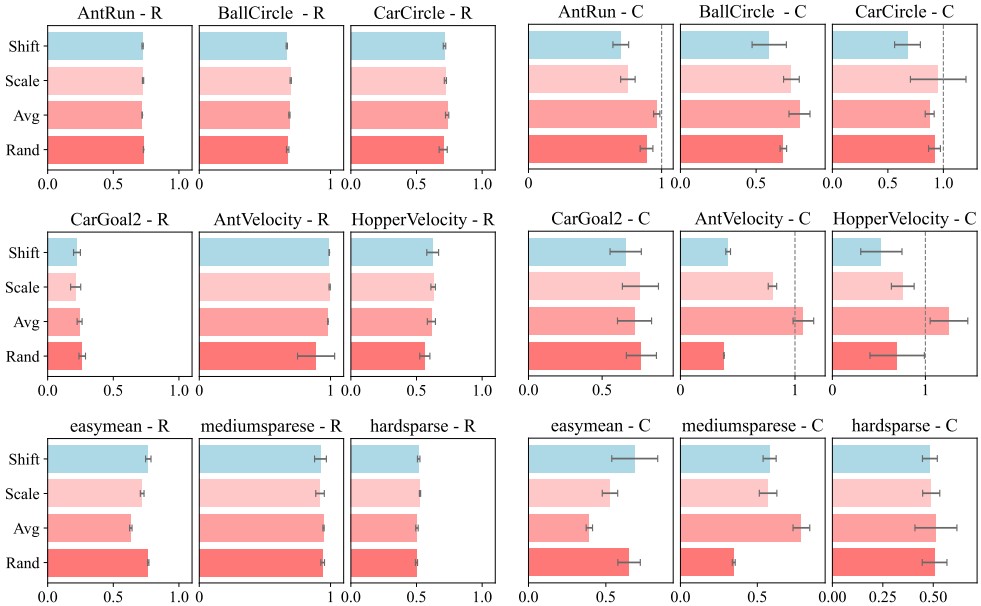

Figure 8: Reward and cost comparison of four CTG realignment strategies across nine representative tasks. Shift achieves the best overall trade-off.

**Experimental Setup.** We evaluate four CTG realignment strategies described above under the B2R, keeping all other components fixed. Experiments are conducted on 9 representative tasks from the DSRL benchmark, selected to cover diverse dynamics and constraint regimes. For each strategy, we report normalized reward, normalized cost, and constraint violation rate, averaged over three random seeds and all constraint levels. This setup allows us to isolate the impact of CTG structure on policy behavior.

**Results and Analysis.** Figure 8 report the reward and cost performance of the four realignment strategies across nine representative tasks.

Overall, the **Shift** strategy achieves the most balanced trade-off: it consistently maintains low cumulative cost while achieving near-optimal reward. This supports the hypothesis that preserving the original cost structure, while globally aligning to the target budget, is sufficient for effective constraint-aware training.

The **Scale** strategy occasionally yields higher reward (e.g., `AntVelocity`), but at the expense of increased cost and more frequent constraint violations, particularly in sparse-cost environments such as `HopperVelocity`. This suggests that proportional rescaling can overstretch cost dynamics and lead to over-optimism in high-return regions.

**Avg** performs reasonably but tends to underperform in tasks with strongly structured or sparse cost signals (e.g., `CarCircle`, `easymean`), likely due to over-smoothing the CTG sequence and erasing important temporal variations.

**Rand** performs surprisingly well across many tasks, maintaining competitive reward and low cost. This suggests that although the reallocation is stochastic, it still respects the underlying cost structure of the environment. As a result, it preserves budget feasibility while introducing minimal structural distortion, making it more robust than Avg in sparse or high-variance environments.

While the four CTG realignment strategies differ in their structural assumptions and empirical performance, all of them outperform the baseline model without realignment. This confirms that the B2R framework itself is robust to variations in cost token design, and that the realignment mechanism—regardless of its specific form—is essential for effective constraint-aware training. These results validate B2R as a general paradigm for aligning training-time supervision with deployment-time constraints. Moreover, the observed differences across strategies suggest that environment-specific design choices can further enhance performance, without undermining the overall value of the framework.

## B.2 Comparison with Recent Offline Safe RL Baselines

To assess the broader applicability of B2R, we compare it against several recent offline safe RL methods, including FAWAC [38], OASIS [35], LSPC [14], and FISOR [38]. These baselines leverage generative modeling or latent constraint inference to improve safety in offline settings.

Table 2: Extended comparison between B2R and recent safe offline RL methods: FISOR, OASIS, LSPC, and FAWAC. Metrics report normalized reward (↑) and cost (↓), averaged over 3 constraint levels, 20 episodes, and 3 seeds. **Bold** indicates safe agents (cost < 1); **Blue** highlights safe agents achieving the highest reward. B2R consistently achieves strong reward while maintaining low cost across diverse tasks.

| Task | FISOR | | OASIS | | LSPC | | FAWAC | | B2R(ours) | |
|---|---|---|---|---|---|---|---|---|---|---|
| | reward↑ | cost↓ | reward↑ | cost↓ | reward↑ | cost↓ | reward↑ | cost↓ | reward↑ | cost↓ |
| BallRun | **0.17** | **0.04** | **0.28** | **0.79** | **0.14** | **0.00** | **0.24** | **0.19** | **0.31** | **0.23** |
| CarRun | **0.85** | **0.15** | **0.85** | **0.02** | **0.97** | **0.13** | **0.97** | **0.13** | **0.96** | **0.08** |
| DroneRun | 0.44 | 2.52 | **0.13** | **0.79** | **0.57** | **0.00** | **0.57** | **0.12** | **0.56** | **0.03** |
| BallCircle | **0.28** | **0.00** | **0.70** | **0.45** | **0.47** | **0.01** | **0.61** | **0.89** | **0.67** | **0.59** |
| CarCircle | **0.24** | **0.15** | **0.76** | **0.89** | **0.72** | **0.04** | **0.42** | **0.63** | **0.71** | **0.68** |
| DroneCircle | **0.49** | **0.02** | **0.60** | **0.25** | **0.58** | **0.60** | **0.57** | **0.79** | **0.57** | **0.34** |
| **Average** | **0.41** | **0.48** | **0.55** | **0.53** | **0.57** | **0.13** | **0.56** | **0.46** | **0.63** | **0.33** |

As shown in Table 2, B2R achieves the highest average reward while maintaining competitive or lower cost across diverse tasks. While some methods (e.g., OASIS, LSPC) perform well in specific

environments, they often suffer from higher cost or limited generalization across constraint levels. In contrast, B2R delivers consistently strong performance without requiring task-specific tuning or architectural modifications. This highlights its robustness and effectiveness as a general-purpose framework. Method details are provided in Section 2.2.

## B.3 Comparison with Buffered CDT Baseline

To verify that the superiority of B2R is not merely a margin effect, we conducted a comparative analysis against a natural baseline: the CDT trained with a stricter buffered cost threshold. This experiment directly evaluates whether simply adjusting the target CTG is sufficient for achieving robust safety, or if B2R's cost realignment mechanism offers a more fundamental advantage. We benchmarked B2R against CDT variants trained with target CTGs set to 70%, 80%, 90%, and 100% of the true constraint threshold $\kappa$.

The results in table 3 reveal the limitations of a simple margin-based baseline. Such an approach often creates an unfavorable trade-off between safety and performance and can be unreliable in complex tasks where no simple buffer value guarantees safety. We identify the root cause in the supervision paradigm: a buffered baseline is still constrained by sparse boundary supervision, learning only from trajectories near a specific cost target. In contrast, B2R's region supervision leverages the entire safe dataset to create a denser learning signal. This confirms that B2R's advantage is not a mere margin effect but stems from a more robust and data-efficient learning paradigm.

Table 3: Comparison of B2R against CDT with buffered cost thresholds. The CDT variants (CDT-X%) are trained with a target CTG set to X% of the true constraint threshold $\kappa$. Metrics report normalized R (↑) and C (↓), averaged over 3 constraint levels, 20 episodes, and 3 seeds. **Bold** indicates safe agents (cost < 1); **Blue** highlights safe agents achieving the highest reward.

| Task | CDT-70% | | CDT-80% | | CDT-90% | | CDT-100% | | B2R | |
|------|-----|-----|-----|-----|-----|-----|-----|-----|-----|-----|
| | R↑ | C↓ | R↑ | C↓ | R↑ | C↓ | R↑ | C↓ | R↑ | C↓ |
| AntVelocity | **0.95** | **0.40** | **0.99** | **0.47** | **0.99** | **0.53** | **0.99** | **0.55** | **0.99** | **0.42** |
| CarCircle | **0.71** | **0.61** | **0.71** | **0.70** | **0.72** | **0.76** | **0.72** | **0.90** | **0.71** | **0.68** |
| mediummean | 0.86 | 1.85 | 0.87 | 1.97 | 0.82 | 1.86 | 0.81 | 1.92 | **0.88** | **0.63** |
| harddense | **0.19** | **0.40** | **0.20** | **0.40** | **0.24** | **0.55** | **0.25** | **0.65** | **0.48** | **0.58** |

## B.4 Performance under Safe Data Scarcity

To quantify the robustness of B2R to the sparsity of safe data, we conducted a data ablation study. We retrained B2R on subsets of the filtered safe dataset, sampled at 5%, 20%, 50%, and 100% of the originally available safe trajectories. The results on four representative tasks are shown in Table 4.

Table 4: B2R performance under data scarcity, with a cost threshold $\kappa = 10$. Metrics report normalized R (↑) and C (↓), averaged over 20 episodes, and 3 seeds. **Bold** indicates safe agents (cost < 1); **Blue** highlights safe agents achieving the highest reward. The policy's performance degrades gracefully, showing resilience even with only 20% of the safe data in many tasks.

| Task | 5% | | 20% | | 50% | | 100% | |
|------|-----|-----|-----|-----|-----|-----|-----|-----|
| | R↑ | C↓ | R↑ | C↓ | R↑ | C↓ | R↑ | C↓ |
| BallCircle | 0.59 | 2.50 | **0.61** | **0.88** | **0.64** | **0.80** | **0.67** | **0.59** |
| DroneCircle | 0.37 | 2.37 | **0.44** | **0.78** | **0.53** | **0.26** | **0.57** | **0.34** |
| easysparse | **0.60** | **0.27** | **0.66** | **0.63** | 0.59 | 0.00 | **0.63** | **0.26** |
| mediummean | **0.35** | **0.33** | 0.43 | 2.57 | **0.42** | **0.09** | **0.75** | **0.06** |

The results demonstrate B2R's robustness. Even with only 20% of the data, B2R maintains safe policies in `BallCircle` and `DroneCircle`. This resilience stems from our region-wide supervision, which creates a denser and more robust learning signal than boundary-focused methods, making our approach less sensitive to data scarcity.

## B.5 Extension to Multiple Constraint Targets

The main body of our work focuses on a single, fixed safety threshold, as this setting mirrors many practical safety-critical applications where reliability under a specific constraint is paramount. However, the B2R framework is flexible and can be extended to handle multiple cost targets within a single model, avoiding the need for retraining for each new constraint.

The methodology for this multi-target extension is as follows:

1. **Multi-Target Data Preparation:** We define a set of target cost thresholds $K = \{k_1, k_2, \dots\}$. For each $k_i \in K$, we filter the original offline dataset to obtain a safe subset $\mathcal{D}_{\text{safe}, k_i}$.

2. **Per-Target CTG Realignment:** For every dataset $\mathcal{D}_{\text{safe}, k_i}$, we apply our CTG-realignment procedure using its specific threshold $k_i$. This yields a set of distinct realigned datasets.

3. **Unified Conditional Training:** We merge all realigned datasets into a single mixed set and train one conditional policy. At inference time, the desired cost threshold $k_i$ is provided as an initial condition to the model.

We conducted an exploratory experiment to validate this approach. Table 5 compares the performance of a single B2R agent trained on a mixed dataset for thresholds $\{10, 20, 40\}$ against specialized agents trained for each single threshold.

The results show that a single multi-target B2R agent can achieve comparable performance to specialized agents across various thresholds. This confirms the promise of this extension, though a deeper investigation into the performance trade-offs remains a topic for future work.

Table 5: Performance comparison between a single multi-target B2R agent and specialized single-target agents. The multi-target agent is trained once on a mixed dataset and evaluated at different target thresholds ($\kappa$). Metrics report normalized R ($\uparrow$) and C ($\downarrow$), averaged over 20 episodes, and 3 seeds. **Bold** indicates safe agents (cost $< 1$); **Blue** highlights safe agents achieving the highest reward. The comparable performance demonstrates the feasibility of extending B2R to handle multiple constraints without retraining.

| Task | $\kappa$ | Multi-Target | | Single-Target | |
|---|---|---|---|---|---|
| | | **R↑** | **C↓** | **R↑** | **C↓** |
| easysparse | 10 | **0.70** | **0.48** | 0.63 | 0.26 |
| | 20 | 0.81 | 1.12 | 0.80 | 1.08 |
| | 40 | **0.86** | **0.70** | **0.89** | **0.81** |
| BallCircle | 10 | **0.68** | **0.90** | 0.65 | 0.81 |
| | 20 | **0.71** | **0.76** | 0.70 | 0.79 |
| | 40 | **0.73** | **0.39** | 0.65 | 0.16 |
| AntVelocity | 10 | **1.00** | **0.73** | 0.98 | 0.45 |
| | 20 | **1.00** | **0.53** | 0.99 | 0.46 |
| | 40 | **0.99** | **0.32** | **1.00** | **0.34** |

## B.6 Comparison with FISOR under Stringent Cost Limits

To specifically evaluate B2R's performance in scenarios with sparse safe data resulting from stringent cost constraints (a "FISOR-style setup"), we conducted a head-to-head comparison with FISOR. Following the protocol from the original FISOR paper, we used tight cost thresholds of $\kappa = 10$ for Safety-Gymnasium tasks and $\kappa = 5$ for others, which are stricter than our main experiments.

The results in Table 6 show that B2R not only remains safe under these stringent conditions but also consistently achieves significantly higher rewards. This suggests that while FISOR's hard-constraint approach can be overly conservative, B2R's region-wide supervision allows it to learn a more effective policy that better balances safety and performance, even when safe data is sparse.

Table 6: Comparison of B2R and FISOR under stringent, low cost-limits. **Bold** indicates safe agents (cost < 1); **Blue** highlights safe agents achieving the highest reward. B2R consistently achieves higher rewards while maintaining robust safety, demonstrating a better safety-performance balance than the more conservative FISOR approach in these settings.

| Task | B2R | | FISOR | |
|---|---|---|---|---|
| | **R↑** | **C↓** | **R↑** | **C↓** |
| BallCircle | **0.60** | **0.98** | 0.34 | 0.00 |
| DroneCircle | **0.53** | **0.58** | 0.48 | 0.00 |
| easysparse | **0.60** | **0.00** | 0.38 | 0.53 |
| medmean | **0.74** | **0.08** | 0.39 | 0.08 |
| harddense | **0.42** | **0.14** | 0.30 | 0.34 |
| AntVelocity | **0.97** | **0.99** | 0.89 | 0.00 |

## C  Implementation Details

### C.1  Inference Algorithm

The following Algorithm 2 provides the pseudocode for the specific implementation of the B2R framework during the inference phase. In this phase, the trained policy is used to make decisions based on the input environment states, while ensuring that safety constraints are respected throughout the process. The goal of this inference phase is to generate a sequence of actions by generating from the trained policy, leveraging both reward and cost information accumulated from past decisions. The process follows a step-by-step approach:

---
**Algorithm 2** Inference with Boundary-to-Region Framework

---
**Require:** Trained policy $\pi_\theta$, constraint threshold $\kappa$, target return $R_0$, initial state $s_0$
**Ensure:** Generated trajectory $\tau = \{(s_t, a_t, r_t, c_t)\}_{t=0}^{H}$
1: Initialize: $\hat{R}_0 \leftarrow R_0, \hat{C}_0 \leftarrow \kappa, s_0 \leftarrow \texttt{env.reset()}, a_{<0} \leftarrow \emptyset$
2: **for** $t = 0$ to $H$ **do**
3:    Construct tokenized context:

$$o_t = [\hat{R}_{t-K:t}, \hat{C}_{t-K:t}, s_{t-K:t}, a_{t-K:t-1}]$$

4:    Predict action: $a_t \sim \pi_\theta(\cdot \mid o_t)$
5:    Execute $a_t$ in environment: $s_{t+1}, r_t, c_t, \texttt{done} \leftarrow \texttt{env.step}(a_t)$
6:    Update cost and return: $\hat{R}_{t+1} \leftarrow \hat{R}_t - r_t, \quad \hat{C}_{t+1} \leftarrow \hat{C}_t - c_t$
7:    **if** done **then**
8:       **break**
9:    **end if**
10: **end for**
11: **return** $\tau = \{(s_t, a_t, r_t, c_t)\}_{t=0}^{t_{\text{final}}}$

---

### C.2  Training Environment

Our experiments utilize the DSRL benchmark [25], which consists of 38 diverse sequential decision-making tasks covering robotic control, navigation, and autonomous driving. These tasks are specifically chosen to challenge agents in a variety of realistic, safety-critical environments. They are drawn from several well-established benchmarks that aim to push the limits of reinforcement learning in both safety and performance. These benchmarks include:

**SafetyGymnasium** [29]: A suite of Mujoco-based environments designed for safe reinforcement learning, featuring diverse tasks (Goal, Button, Push, Circle) with adjustable difficulty levels and various safety constraints.

**BulletSafetyGym** [11]: Built on the PyBullet physics engine, this benchmark extends SafetyGymnasium with additional agents (Ball, Car, Drone, Ant) and shorter episode horizons.

**MetaDrive** [22]: A self-driving simulator based on the Panda3D game engine, providing complex road conditions and dynamic traffic interactions to evaluate safe RL in realistic driving scenarios.

### C.3 Evaluation Metrics

We evaluate the performance of B2R using the metrics from the DRSL to ensure fair comparisons across different tasks. For tasks within MetaDrive and BulletSafetyGym, we set the cost thresholds to 10, 20, and 40, ensuring a balanced trade-off between performance and constraint adherence. In contrast, for environments in SafetyGymnasium, the cost thresholds are higher, specifically 20, 40, and 80, to account for the different cost dynamics and safety requirements in these settings.

**Normalized Reward**  The normalized reward measures policy performance and is computed as:

$$R_{\text{normalized}} = \frac{R_\pi - r_{\min}(\mathcal{M})}{r_{\max}(\mathcal{M}) - r_{\min}(\mathcal{M})} \times 100, \tag{42}$$

where $R_\pi$ is the evaluated reward return, and $r_{\max}(\mathcal{M})$ and $r_{\min}(\mathcal{M})$ are the empirical maximum and minimum rewards for task $\mathcal{M}$.

**Normalized Cost**  The normalized cost is defined as:

$$C_{\text{normalized}} = \frac{C_\pi + \epsilon}{\kappa + \epsilon}, \tag{43}$$

where $C_\pi$ is the evaluated cost return, $\kappa$ is the target threshold, and $\epsilon$ is a small positive number ensuring numerical stability.

### C.4 Baseline Methods

To ensure a comprehensive evaluation, we compare our approach against several state-of-the-art offline safe reinforcement learning algorithms. These baselines encompass a range of methodologies, including behavior cloning, sequential modeling, distribution correction, and Lagrangian-based approaches. Some results are directly obtained from the DSRL benchmark, while others are reproduced using their official implementations.

- **BC-All / BC-Safe**: Behavior Cloning (BC) serves as a fundamental baseline by training policies to mimic expert demonstrations. BC-All utilizes the entire dataset, whereas BC-Safe exclusively uses safe trajectories to ensure policy compliance with constraints.
- **CDT (Conditional Decision Transformer)** [26]: A Decision Transformer-based approach that incorporates safety constraints by conditioning on cost-related tokens, allowing the model to learn safe policies in an autoregressive manner.
- **BCQ-Lag** [8]: A Lagrangian-based extension of BCQ that penalizes unsafe actions while optimizing for reward. The Lagrangian multiplier dynamically adjusts to enforce safety constraints.
- **CPQ (Constraint-Penalized Q-learning)** [34]: Treats out-of-distribution actions as unsafe and penalizes them by modifying the Q-value function, preventing policy optimization on unsafe state-action pairs.
- **COptiDICE** [19]: A distribution correction estimation (DICE)-based offline safe RL method that extends OptiDICE [18], explicitly enforcing cost constraints while optimizing policy performance.
- **TraC (Trajectory-Constrained RL)** [10]: A trajectory-based approach that incorporates cost-awareness into safe RL by guiding the policy learning process with explicit constraints.

### C.5 Hyperparameter Settings

All models are trained for **20 epochs**, each consisting of **5000 gradient steps**, totaling **100,000 training steps**. To evaluate robustness, we use **three random seeds** for all experiments. Environment-specific cost thresholds are listed in Table 7, along with other key hyperparameters.

Table 7: Hyperparameter settings for B2R experiments.

| Category | Hyperparameter | Value |
|---|---|---|
| **Optimizer** | Optimizer Type | Lamb |
| | Learning Rate | 0.0001 |
| | Batch Size | 2048 |
| | Gradient Clipping | 0.25 |
| **Training Strategy** | Steps per Epoch | 5000 |
| | Total Epochs | 20 |
| | Early Stopping | Reward stagnation |
| **Transformer Model** | Hidden Dimension | 128 |
| | Attention Heads | 8 |
| | Transformer Layers | 3 |
| | Dropout | 0.1 |
| **Environment-Specific Settings** | Context Length (MetaDrive) | 3 |
| | Context Length (Others) | 10 |

### C.6 Computing resources.

The experiments are conducted on a Linux-based server equipped with an Intel Core i9-14900K 32-Core Processor, one NVIDIA GeForce RTX 4070 GPU, and 64 GB of RAM. The implementation is based on PyTorch (v1.13.1) with CUDA 12.4. Training is performed for 100K steps, with gradient updates running on a single GPU. The total training time per task averages approximately 2 hours, which is comparable to CDT, another Transformer-based method. Evaluation is conducted separately on CPU/GPU to ensure efficiency.

