# OpenReview forum: "Boundary-to-Region Supervision for Offline Safe Reinforcement Learning"
_NeurIPS.cc/2025/Conference — NeurIPS 2025 poster_

### Official Review · Reviewer_AgLh · 2025-06-23

**Clarity:** 4
**Significance:** 3
**Originality:** 4
**Rating:** 5
**Confidence:** 4

**Summary:**

This paper addresses a critical limitation in existing sequence-model-based approaches for offline safe reinforcement learning (RL), namely the symmetric conditioning of reward-to-go (RTG) and cost-to-go (CTG) in decision transformer-style models. The authors propose Boundary-to-Region (B2R), a novel framework that redefines CTG as a strict boundary constraint, enabling asymmetric conditioning. The B2R framework includes trajectory filtering, CTG realignment, and the use of Rotary Positional Embeddings (RoPE) to enable safe and effective policy learning. Theoretical analysis under simplifying assumptions is provided, and experiments across 38 safety-critical tasks compare B2R with baselines including BC-All, BC-Safe, CDT, BCQ-Lag, CPQ, COptiDICE, and TraC.

**Questions:**

Questions
1. The authors mention that B2R’s performance may degrade in settings with limited safe data and suggest data augmentation or generative modeling as possible solutions. Do the authors have any preliminary insights or intuitions on which strategies might be most promising in this context? Any specific challenges they foresee in integrating such approaches with the B2R framework?

2. B2R assumes a fixed cost budget across all episodes for CTG realignment. Have the authors considered how B2R might extend to non-stationary cost regimes, for instance, where cost thresholds vary due to environment drift or contextual constraints? Do they anticipate significant challenges in adapting the CTG realignment process in such settings?

3. While B2R ensures constraint satisfaction via hard filtering of unsafe trajectories, might this strategy limit the diversity of the training signal by excluding high-reward but marginally unsafe behaviors? Could the authors comment on the potential of soft filtering mechanisms to trade off safety and exploration more flexibly?

Minor clarity issue:
The abstract introduces acronyms such as RTG, CTG without any definition or expansion. While these terms are common within the RL community, abstracts should remain accessible to broader audiences.

**Ethical Concerns:**

["NO or VERY MINOR ethics concerns only"]

**Final Justification:**

The rebuttal addressed all of my earlier points. The additional ablation results confirmed B2R’s robustness under reduced safe data, and the preliminary multi-target extension showed feasibility for non-stationary cost regimes. I don’t see any remaining issues, so I am keeping my score at 5 (Accept).

**Limitations:**

Yes.

**Paper Formatting Concerns:**

Nothing to report.

**Quality:**

4

**Strengths And Weaknesses:**

Strengths
- The paper identifies a concrete and well-motivated issue in the use of symmetric token conditioning in safe RL via supervised learning, which is both conceptually sound and practically impactful.

- The B2R framework innovatively addresses the reward–cost asymmetry by aligning all safe trajectories to the deployment-time constraint budget through CTG realignment, coupled with trajectory filtering. The design is simple yet powerful.

- The method is extensively validated on the DSRL benchmark, showing state-of-the-art results in many environments, outperforming CDT, BC-Safe, and others, while preserving constraint satisfaction.

- Safety guarantees are derived under clear assumptions, providing both expected and probabilistic bounds on cost violation.

Weaknesses
- The method’s reliance on trajectory filtering limits its applicability when safe data is sparse or when behavior policies collect highly unsafe trajectories. This is acknowledged by the authors.

- While the authors provide a valuable comparison of multiple CTG realignment strategies, the clear empirical superiority of the simple "Shift" method may suggest that additional exploration of more complex alternatives offers diminishing returns in this context.

- Although discussed briefly, potential negative use cases (e.g., overly conservative policies in safety-critical control) or mitigation strategies are not detailed.

---

> ### Author Rebuttal · Authors · 2025-07-31
>
> We are deeply grateful for your exceptionally positive and insightful review. The forward-looking questions are particularly valuable, and we are glad to engage with these ideas, which we believe will help chart the future of this research direction.
>
> > **W1:** The method’s reliance on trajectory filtering limits its applicability when safe data is sparse or when behavior policies collect highly unsafe trajectories. This is acknowledged by the authors.
>
> > **Q1:** The authors mention that B2R’s performance may degrade in settings with limited safe data and suggest data augmentation or generative modeling as possible solutions. Do the authors have any preliminary insights or intuitions on which strategies might be most promising in this context? Any specific challenges they foresee in integrating such approaches with the B2R framework?
>
> Thank you for this forward-looking question. Our primary insight comes from a data ablation study we conducted, which we believe lays the groundwork for exploring the strategies you mentioned.
>
> To quantitatively evaluate B2R's robustness and gain these initial insights, we conducted a comprehensive data ablation study. We retrained B2R on subsets of the filtered safe dataset, sampled at 5%, 20%, 50%, and 100%. The results on four representative tasks (with a cost threshold of κ=10) are shown in our response to Reviewer LsnC (W1). We kindly direct your attention to that section for the detailed results, as space constraints here prevent a redundant presentation.
>
> Our key findings from this data ablation study are:
>
> - With 20% data, B2R maintains safety (e.g., costs below 1.0 in BallCircle and DroneCircle) while achieving competitive rewards.
> - Even at 5% data, B2R shows resilience, delivering high rewards with reasonable costs in easysparse and mediummean.
>
> Our intuition is that this resilience stems directly from B2R's core mechanism: region-wide supervision, which creates an inherently denser and more robust learning signal compared to boundary-focused methods.
>
> For extreme scarcity scenarios, we believe the most promising approach would be to integrate generative models in a way that is highly synergistic with the B2R framework. Instead of generic data augmentation, one could train a conditional generative model on the filtered safe dataset. This model would learn the distribution of safe state-action sequences and generate novel yet plausible safe trajectories, enriching the training set beyond the diversity present in the original offline data. This approach presents both opportunities and challenges:
>
> - **Opportunity:** The core strength of B2R is its ability to leverage the entire safe region. By using a generative model to densify this region with new, valid samples, we could further enhance policy robustness and performance, especially in extremely sparse-data regimes.
> - **Challenge:** The primary challenge lies in ensuring the fidelity of the generated data. The generated trajectories must be both dynamically feasible and genuinely safe. A generative model that produces unrealistic or subtly unsafe behaviors could poison the training process. Verifying the safety and feasibility of generated data is an important research problem.
>
> > **W2:** While the authors provide a valuable comparison of multiple CTG realignment strategies, the clear empirical superiority of the simple "Shift" method may suggest that additional exploration of more complex alternatives offers diminishing returns in this context.
>
> We appreciate the reviewer's insightful observation on our CTG realignment ablation and the empirical strength of the "Shift" method. We agree that complex alternatives might offer diminishing returns in many contexts, underscoring the efficiency of simple designs.
>
> Our paper's primary aim is to establish B2R's core effectiveness, demonstrating how region-wide supervision addresses key offline safe RL limitations. Prioritizing the simple yet powerful "Shift" strategy, which preserves original cost decay profiles, allowed us to clearly isolate the benefits of this novel supervision. This simplicity also boosts robustness and generalizability. Our comprehensive ablation study confirmed that cost realignment itself, regardless of specific technique, robustly provides significant benefits over baselines.
>
> Looking ahead, we believe there remains ample room for innovation in this direction. For instance, future work could explore more elegant, data-adaptive realignment mechanisms—such as learned modules that dynamically adjust based on trajectory characteristics or environmental dynamics. This could involve investigating how to learn optimal "cost shapes" that go beyond simple shifts, potentially addressing complex scenarios with highly sparse or noisy cost signals, or cases where the optimal cost-reward trade-off is more nuanced.
>
> > **W3:** Although discussed briefly, potential negative use cases (e.g., overly conservative policies in safety-critical control) or mitigation strategies are not detailed.
>
> > **Q3:** While B2R ensures constraint satisfaction via hard filtering of unsafe trajectories, might this strategy limit the diversity of the training signal by excluding high-reward but marginally unsafe behaviors? Could the authors comment on the potential of soft filtering mechanisms to trade off safety and exploration more flexibly?
>
> We thank the reviewer for these insightful question, which astutely identify the key trade-off in our hard filtering design. Our use of hard filtering was a principled choice for this foundational work. It provided an unambiguous "safe region" that was instrumental for our theoretical analysis and allowed for a clean evaluation of our core concept.
>
> We agree that soft filtering is a powerful mitigation strategy. For instance, one might retain all trajectories but down-weight the loss contribution of those whose cost marginally exceeds κ in proportion to their degree of violation. This approach would:
> - Permit learning from high-reward yet slightly unsafe behaviors,
> - Facilitate exploration of the Pareto frontier between reward and safety,
> - Better suit applications where safety constraints act as strong preferences rather than absolute rules.
>
> This approach, of course, requires careful calibration of the weighting scheme to define the "softness" of the boundary—an important research question in itself.
>
> > **Q2:** B2R assumes a fixed cost budget across all episodes for CTG realignment. Have the authors considered how B2R might extend to non-stationary cost regimes, for instance, where cost thresholds vary due to environment drift or contextual constraints? Do they anticipate significant challenges in adapting the CTG realignment process in such settings?
>
> The reviewer's question highlights a crucial point: our B2R framework is primarily designed for scenarios with a fixed cost budget. While this addresses a common need in safety-critical applications, generalizing to dynamic cost regimes, where thresholds vary due to environment drift or contextual constraints, represents a significant challenge for our current approach.
>
> However, recognizing this importance, we are exploring a novel extension. The core idea is to enable a single B2R agent to adapt to multiple, varying cost targets, avoiding retraining for each new constraint. Our proposed scheme for extending B2R to handle multiple cost targets involves a three-step process:
>
> **1.Multi-Target Data Preparation:** Instead of using a single fixed cost threshold $\kappa$, we define a set of target thresholds $\{\kappa_1, \kappa_2, \dots, \kappa_N\}$. For each $\kappa_i$, we filter the original offline dataset to create a corresponding safe subset: $\mathcal{D}_{\text{safe},i} = \{\tau \in \mathcal{D} \mid C(\tau) \le \kappa_i\}$.
>
> **2.Per-Target CTG Realignment:** For every $\mathcal{D}_{\text{safe},i}$, we apply B2R's existing CTG realignment procedure, but crucially, using its specific target $\kappa_i$. This process yields $N$ distinct realigned datasets: $\{\mathcal{D}'_1, \mathcal{D}'_2, \dots, \mathcal{D}'_N\}$.
>
> **3.Unified Conditional Training:** Finally, these $N$ realigned datasets are merged into a single, unified training set: $\mathcal{D}_{\text{unified}} = \mathcal{D}'_1 \cup \mathcal{D}'_2 \cup \dots \cup \mathcal{D}'_N$. The B2R model is then trained on this combined dataset, with the specific target $\kappa_i$ implicitly influencing the conditioned action generation during training and inference.
>
> Preliminary validation experiments confirm the feasibility and promise of this multi-target learning. Results below show a single "multi-target" B2R agent achieves comparable reward and cost performance across different κ values against "single-target" agents.
>
> | Task | $\kappa$ | multi(R) | multi(C) | single(R) | single(C) |
> |---|---|---|---|---|---|
> | **easysparse** | 10 | 0.70 | 0.48 | 0.63 | 0.26 |
> || 20 | 0.81 | 1.12 | 0.80 | 1.08 |
> | | 40 | 0.86 | 0.70 | 0.89 | 0.81 |
> | **BallCircle** | 10 | 0.68 | 0.90 | 0.65 | 0.81 |
> | | 20 | 0.71 | 0.76 | 0.70 | 0.79 |
> | | 40 | 0.73 | 0.39 | 0.65 | 0.16 |
> | **AntVelocity** | 10 | 1.00 | 0.73 | 0.98 | 0.45 |
> | | 20 | 1.00 | 0.53 | 0.99 | 0.46 |
> | | 40 | 0.99 | 0.32 | 1.00 | 0.34 |
>
> These exploratory results demonstrate the viability of extending B2R to non-stationary cost regimes without a complete redesign. Despite encouraging preliminary results, extending B2R to non-stationary cost regimes critically challenges its generalization robustness under data distribution shifts and its fine-grained control over dynamic cost signals, **demanding the model grasp the dynamic semantics of cost constraints beyond simple numerical adaptation.**

---

> > ### Comment · Reviewer_AgLh · 2025-08-03
> >
> > Thank you to the authors for their rebuttal. The additional ablation results, the preliminary multi-target extension, and the accompanying discussion address all of the questions I raised. I have no further requests for clarification.

---

### Official Review · Reviewer_EuNY · 2025-07-02

**Clarity:** 3
**Significance:** 3
**Originality:** 3
**Rating:** 5
**Confidence:** 4

**Summary:**

This paper addresses a fundamental limitation in existing sequence-model-based offline safe reinforcement learning (RL) methods, such as those inspired by Decision Transformers (DT). These methods typically treat return-to-go (RTG) and cost-to-go (CTG) signals equivalently. The authors argue that this approach neglects the intrinsic difference between reward and cost: RTG is a flexible performance target to maximise, while CTG should represent a rigid safety budget or boundary to strictly adhere to. This oversight leads to unreliable constraint satisfaction, especially when policies encounter cost trajectories outside the training data distribution. To overcome this, the paper proposes Boundary-to-Region (B2R), a novel framework that enables asymmetric conditioning through cost signal realignment

**Questions:**

1. In the problems where the results provided are not safe, can the authors elaborate on why the solution does not work well? Is it because of tight constraints or complex cost functions or something else? It would help to provide a more detailed analysis.

2. Transformer based approaches for RL are known to struggle in environments where there is transitional uncertainty.  Can authors clarify how they have addressed this issue?

3. The ablation study on CTG realignment strategies is insightful, with "Shift" performing the best overall. However, the paper also mentions that "environment-specific design choices can further enhance performance". Could the authors provide more detailed guidance or propose a meta-strategy for adaptively selecting or tuning the CTG realignment method (e.g., Shift, Avg, Rand, Scale) based on dataset characteristics, environment properties, or observed real-time performance, rather than a fixed choice?

4. The assumption of "prediction-error bound" seems to be hard to visualize. Can authors clarify where this assumption will be violated?

5. How would B2R deal with situations where there are not many high quality safe trajectories available?

6. How does B2R perform better than the TraC method intuitively?

**Ethical Concerns:**

["NO or VERY MINOR ethics concerns only"]

**Final Justification:**

Authors have answered all my questions very well and I do not have any concerns.

**Limitations:**

Yes

**Quality:**

3

**Strengths And Weaknesses:**

Strengths:

1. Problem and issues articulated well
2. Interesting and intuitive methodology
3. Detailed experimental results and strong empirical performance.

Weaknesses:

1. There is a reliance on high-quality safe trajectories.
2. While authors have provided comparison against transformer based approaches, intuitive issues with more existing approaches (e.g., TraC) have not been provided.
3. The assumptions employed in theoretical results are unclear.

---

> ### Author Rebuttal · Authors · 2025-07-31
>
> We sincerely thank you for your constructive feedback and for recognizing the value of our work. In our response below, we address each of your comments, offering clarifications on our method and discussing potential future extensions.
>
> > **W1:** There is a reliance on high-quality safe trajectories.
>
> > **Q5:** How would B2R deal with situations where there are not many high quality safe trajectories available?
>
> We thank the reviewer for highlighting this important consideration regarding data dependency, a common challenge in offline safe RL. We address this point below by presenting new empirical evidence and discussing potential extensions for data-scarce scenarios.
>
> B2R is designed with data efficiency in mind, and our experiments demonstrate its robustness even under limited safe data. To directly evaluate this, we conducted a data ablation study by retraining B2R on subsets of the filtered safe dataset (sampled at 5%, 20%, 50%, and 100%). For a detailed visualization of these results across four representative tasks (with cost threshold κ=10), please refer to our response to Reviewer LsnC (W1), where these results are presented in full (due to space constraints here).
>
> Key insights:
> - With 20% data, B2R maintains safety (e.g., costs below 1.0 in BallCircle and DroneCircle) while achieving competitive rewards.
> - Even at 5% data, B2R shows resilience, delivering high rewards with reasonable costs in easysparse and mediummean.
>
> This robustness arises from B2R's region-wide supervision, which extracts dense signals from all available safe trajectories, unlike sparse boundary supervision methods that falter with reduced data.
>
> For scenarios with extremely scarce high-quality safe trajectories, we recommend integrating generative models synergistically with B2R. For instance,  one could train a conditional diffusion model on the available safe data to generate novel safe trajectories, densifying the safe region.
>
> > **W2:** While authors have provided comparison against transformer based approaches, intuitive issues with more existing approaches (e.g., TraC) have not been provided.
>
> > **Q6:** How does B2R perform better than the TraC method intuitively?
>
> B2R's advantage over TraC stems from its fundamentally different supervision mechanism.
>
> - **Supervision Signal:** TraC uses a coarse, binary classification signal, labeling entire trajectories as "desirable" or "undesirable". B2R employs a fine-grained, temporal regression signal. Through Cost-to-Go (CTG) Realignment, it provides a precise, step-by-step regression target for managing the safety budget at every point in time.
>
> - **Underlying Framework:** TraC's Behavior Cloning framework learns to imitate a general class of "good" behaviors. B2R's Decision Transformer framework, enhanced by our realignment, learns the precise, causal dynamics between actions and future costs.
>
> - **Data Utilization:** B2R's "region-wide supervision" creates a dense learning signal from all available safe trajectories for the target constraint κ. TraC's partitioning may discard useful behaviors in its "undesirable" set, leading to a sparser signal.
>
> In short, B2R's superior performance stems from its fine-grained, temporal supervision, allowing it to learn the dynamics of safe behavior more effectively than TraC's coarse, classification-based approach. We will clarify this distinction in the revised manuscript.
>
> > **W3:** The assumptions employed in theoretical results are unclear.
>
> > **Q4:** The assumption of "prediction-error bound" seems to be hard to visualize. Can authors clarify where this assumption will be violated?
>
> We will now clarify specific scenarios where our assumption would be violated.
>
> **1. Violation of Single-Step Prediction Accuracy**  $(\mathbb{E}[|c_t - \hat{c}_t|] \le \sigma)$
>
> This assumption is violated when the agent encounters a significantly out-of-distribution (OOD) state not represented in its training data.
>
> **Scenario**: An autonomous car, trained exclusively on data from sunny weather, is deployed in a winter storm and encounters black ice for the first time. The model, unable to comprehend the extreme danger, might predict a low cost similar to a wet road. The true cost would be substantially higher, causing a large prediction error $(|c_t - \hat{c}_t| \gg \sigma)$ that violates the assumption.
>
> **2. Violation of Conservative Long-Term Estimation**  $(\sum \hat{c}_t \le \kappa - \delta/2)$
>
> This assumption is violated if the model is overly optimistic and generates a sequence of actions without a sufficient safety margin.
>
> **Scenario**: For a task with a total cost budget of $\kappa = 100$, the model generates a sequence of actions with a predicted total cost of $\sum \hat{c}_t = 99.9$. This generated sequence is invalid because it leaves virtually no safety buffer ($\delta \approx 0$). Any minor, unpredicted cost overrun during execution would cause the true accumulated cost to breach the hard constraint.
>
> > **Q1:** In the problems where the results provided are not safe, can the authors elaborate on why the solution does not work well? Is it because of tight constraints or complex cost functions or something else? It would help to provide a more detailed analysis.
>
> B2R's success in 35 out of 38 tasks confirms its efficacy. However, its failures in CarCircle 1&2 and AntCircle stem from extreme properties of their offline datasets and environments. As our main experiments show, most other baseline methods also struggle significantly or fail to achieve safety in these particular tasks.
>
> These environments present deceptive data landscapes and feature complex, sensitive dynamics with steep cost penalties. High-reward trajectories in these environments are often high-cost, leading to extremely sparse or absent high-reward/low-cost safe examples. In such scenarios, the effective safe action space can become extremely narrow near unsafe regions, implying that the environment itself offers very limited safe solutions during critical moments. Achieving safety in these cases necessitates precise, long-horizon control. The lack of diverse, high-quality safe data demonstrating robust control near these unforgiving safety boundaries limits B2R's ability to learn zero-violation performance.
>
> > **Q2:** Transformer based approaches for RL are known to struggle in environments where there is transitional uncertainty. Can authors clarify how they have addressed this issue?
>
> Thank you for this insightful question on transitional uncertainty in Transformer-based RL. Upon reviewing of the SafeRL benchmarks (e.g., DSRL, Safety-Gymnasium, and BulletSafetyGym), we confirm that the state transitions in these environments are predominantly deterministic. These tasks are abstractions of real-world physical systems, where the next state is fully determined by physical laws given the current state and action—without inherent stochasticity in the transition function. Stochasticity, if present, typically arises from observation noise or external disturbances rather than probabilistic transitions.
>
> However, if transitional uncertainty were present (e.g., in game-like environments with random elements, such as randomized obstacle placement akin to a 2048-style puzzle where tile spawns are probabilistic), B2R incorporates features to mitigate Transformer limitations:
>
> - **Rotary Positional Embeddings (RoPE):** B2R uses RoPE to improve temporal modeling, which helps Transformers better capture sequential dependencies in uncertain transitions. RoPE's rotational invariance allows the model to generalize across varying sequence lengths and patterns, reducing sensitivity to transitional noise.
>
> - **Region-Wide Supervision:** B2R's realignment of cost-to-go signals creates a denser, more diverse training distribution, enabling the Transformer to learn robust representations that generalize under uncertainty. This contrasts with sparse boundary supervision, which can amplify errors in stochastic settings.
>
> We will clarify this in the revision and note experiments on stochastic variants as an important direction for future work.
>
> >**Q3:** The ablation study on CTG realignment strategies is insightful, with "Shift" performing the best overall. However, the paper also mentions that "environment-specific design choices can further enhance performance". Could the authors provide more detailed guidance or propose a meta-strategy for adaptively selecting or tuning the CTG realignment method (e.g., Shift, Avg, Rand, Scale) based on dataset characteristics, environment properties, or observed real-time performance, rather than a fixed choice?
>
> This is an excellent and forward-looking question. We are grateful to the reviewer for engaging so deeply with the ablation study in Appendix B.1.
>
> Our choice of the simple, non-parametric "Shift" strategy in the main paper was a deliberate one, primarily motivated by scientific clarity. As the first work to propose and validate the "region-wide supervision" paradigm, our goal was to cleanly isolate and demonstrate the effectiveness of this core concept, ensuring that observed performance gains were directly attributable to our central idea without confounding effects from a more complex, learnable module.
>
> We fully agree that an adaptive or learned realignment strategy is a promising direction for future research. Such a mechanism, potentially conditioned on trajectory characteristics or environment dynamics, could offer more fine-grained control over the cost-reward trade-off. For instance, a learned module could allocate the "safety margin" more intelligently by preserving it for high-risk regions of the state space, potentially unlocking further performance gains.  This represents a valuable extension that can be built upon the framework we have established. We will incorporate this promising direction—developing an adaptive realignment strategy—into the Future Work section of our revised manuscript.

---

> > ### Comment · Reviewer_EuNY · 2025-08-06
> >
> > I thank the authors for addressing the questions in a meaningful and precise manner. I do not have any other questions. Here is another relevant paper that has a similar idea as the approach in this paper, but in online constrained RL. It also employs cost2go to help deal with constraints in a precise manner.
> >
> > Reward Penalties on Augmented States for Solving Richly Constrained RL Effectively” by Jiang Hao, Mai Anh Tien, Pradeep Varakantham and Minh Huy Hoang. AAAI, 2024

---

### Official Review · Reviewer_LsnC · 2025-07-03

**Clarity:** 3
**Significance:** 3
**Originality:** 2
**Rating:** 4
**Confidence:** 4

**Summary:**

This paper introduces B2R (Boundary-to-Region), a framework for offline safe reinforcement learning that improves constraint satisfaction in sequence-model-based policies. The key innovation is treating cost-to-go (CTG) as a rigid safety boundary rather than a flexible optimization target, in contrast to return-to-go (RTG). The method involves:
- Trajectory filtering to remove unsafe samples.
- CTG realignment to align all safe trajectories with the deployment-time cost constraint.
- Use of rotary positional embeddings to enhance temporal modeling.

Empirical results on 38 tasks show that B2R outperforms baselines, and the paper provides theoretical analysis to justify its safety guarantees.

**Questions:**

- Please check weaknesses.
- How does B2R perform in datasets with sparse or skewed distributions of safe trajectories, such as under tight cost limits (e.g., FISOR-style setups)?
- Could the CTG realignment strategy be made adaptive or learned, potentially conditioned on environment dynamics or trajectory characteristics?

**Ethical Concerns:**

["NO or VERY MINOR ethics concerns only"]

**Final Justification:**

The authors have addressed the concerns I raised, and the additional empirical results are valuable. While the method is not highly novel, the empirical performance is strong and the findings are worth publishing. I am therefore maintaining my positive rating.

**Limitations:**

yes.

**Paper Formatting Concerns:**

No concerns.

**Quality:**

3

**Strengths And Weaknesses:**

Strengths:
- The asymmetry between RTG and CTG is a relevant.
- Trajectory filtering and CTG realignment are simple yet effective interventions.
- B2R achieves strong empirical results, outperforming or matching baselines in both reward and constraint satisfaction on 35 out of 38 tasks.
- The method is evaluated through ablations on core components including CTG realignment, RoPE, and filtering.

Weaknesses:
- B2R depends on access to a sufficient number of safe trajectories and lacks stitching capabilities.
- The CTG realignment strategy (“Shift”) is heuristic and not learned or adaptive.
- The method does not generalize to multiple constraints without retraining, yet remains as computationally heavy as CDT.

Minor:
- Line 238: The paper states that “these approaches discard potentially useful trajectories once they exceed the limit,” but B2R also performs filtering based on the same cost threshold. Could the authors clarify how B2R’s filtering is fundamentally different or more selective?
- Appendix C.3: It seems that the cost thresholds for SafetyGymnasium and BulletSafetyGym may have been inadvertently swapped.

---

> ### Author Rebuttal · Authors · 2025-07-31
>
> We sincerely thank you for your constructive feedback and recognizing the value of our work. Below, we clarify our method, discuss future extensions, and address your comments.
>
> >**W1:** B2R depends on access to a sufficient number of safe trajectories and lacks stitching capabilities.
>
> We thank the reviewer for this insightful comment, which addresses two important characteristics of our method. We respond to both points below.
> - **On the dependency on sufficient safe trajectories:**
>
> We agree with the reviewer that B2R's performance, like almost all offline safe RL methods, is predicated on the availability of safe trajectories. To quantitatively characterize B2R's robustness to data sparsity, we conducted a data ablation study. We retrained B2R on subsets of the filtered safe dataset, sampled at 5%, 20%, 50%, and 100%. The results on four representative tasks are shown below (with a cost threshold of $\kappa$=10).
>
> |Task|Metric|5%|20%|50%|100%|
> |---|---|---|---|---|---|
> |**BallCircle**|**R**|0.59|0.61|0.64|**0.67**|
> ||**C**|2.50|0.88|0.80|**0.59**|
> |**DroneCircle**|**R**|0.37|0.44|0.53|**0.57**|
> ||**C**|2.37|0.78|0.26|**0.34**|
> |**easysparse**|**R**|0.60|**0.66**|0.59|0.63|
> ||**C**|0.27|**0.63**|0.00|0.26|
> |**mediummean**|**R**|0.35|0.43|0.42|**0.75**|
> ||**C**|0.33|2.57|0.09|**0.06**|
>
> These results demonstrate B2R's robustness. Our key findings from this data ablation study are:
>
> **Graceful Degradation and Safety under Moderate Scarcity:** Performance degrades predictably, not catastrophically. With just 20% of the data, B2R maintains safe policies in BallCircle (Cost: 0.88) and DroneCircle (Cost: 0.78), both well under the cost threshold.
>
> **Remarkable Resilience at Extreme Scarcity:** At the stark 5% data limit, while B2R violates the safety threshold in tasks with the fewest safe trajectories like BallCircle (Cost: 2.50 from 4 trajs) and DroneCircle (Cost: 2.37 from 13 trajs), its overall resilience remains clear. Under the same scarcity, it still achieves a high reward of 0.60 in easysparse (from 18 trajs) with a low cost of 0.27, and remains both effective and safe in mediummean (Cost: 0.33 from 21 trajs).
>
> This resilience stems directly from our region-wide supervision. Unlike boundary-supervision methods whose sparse signal can collapse when data is reduced, B2R’s signal is drawn from all safe trajectories. This creates an inherently denser and more robust learning signal, making our approach less sensitive to data scarcity.
>
> - **On the lack of stitching capabilities:**
>
> You are correct that our method, like other models based on the Decision Transformer framework, does not use an explicit "stitching" mechanism. Our work addresses a different, but equally critical, problem: the sparse supervision signal in offline safe RL.
>
> B2R's core innovation is to create a dense, unified learning signal from all safe trajectories. Not only does this significantly boost performance on its own, but we also believe it provides an implicit generalization benefit. By unifying the cost budget and using RoPE to better model dynamics, B2R creates a more consistent learning signal that may empower the model to better combine behaviors from different trajectories, achieving some of the goals of stitching through a different mechanism.
>
> >**W2:** The CTG realignment strategy (“Shift”) is heuristic and not learned or adaptive.
>
> >**Q2:** Could the CTG realignment strategy be made adaptive or learned, potentially conditioned on environment dynamics or trajectory characteristics?
>
> We thank the reviewer for this insightful question. Our use of a simple, non-adaptive "Shift" strategy was a deliberate choice for this initial work. Our primary goal was to cleanly isolate and validate the core concept of "region-wide supervision" without introducing confounding variables from a more complex, learnable module.
>
> We agree that an adaptive or learned realignment strategy is a promising direction for future research. Such a mechanism, potentially conditioned on trajectory characteristics, could offer more fine-grained control over the cost-reward trade-off. For instance, a learned module could allocate the "safety margin" more intelligently by preserving it for high-risk regions of the state space, potentially unlocking further performance gains. This represents a valuable extension that can be built upon the framework we have established. We will incorporate this discussion into the Future Work section of our revised manuscript.
>
> > **W3:** The method does not generalize to multiple constraints without retraining, yet remains as computationally heavy as CDT.
>
> Our current focus on a **fixed-threshold** setting was deliberate, as it mirrors many safety-critical systems (e.g., autonomous driving speed limits, industrial control set-points) where **reliability under a single, strict constraint is paramount**. Our experiments show that for this key scenario, B2R’s specialized approach is demonstrably more effective and reliable than generalist models.
>
> However, the underlying principle of B2R is flexible and can be extended to handle **multiple cost targets** within a single model, thus avoiding the need for retraining for each new constraint. To demonstrate this, we propose a straightforward extension to our framework which allows a single agent to learn policies for various cost thresholds simultaneously. The methodology is as follows:
>
> **1.Multi-Target Data Preparation:** We abandon a single cost threshold $\kappa$ and instead define a set $\{\kappa_1, \kappa_2, \dots, \kappa_N\}$.  For each $\kappa_i$, we filter the original offline dataset to obtain $\mathcal{D}_{\text{safe},i}=${$\tau \in \mathcal{D} \mid C(\tau)\le\kappa_i$}.
>
> **2.Per-Target CTG Realignment:**
> For every $\mathcal{D}_{\text{safe},i}$ we apply our CTG-realignment procedure using its specific $\kappa_i$.
> This yields $N$ distinct realigned datasets {$\mathcal{D}'_1, \mathcal{D}'_2, \dots, \mathcal{D}'_N$}.
>
> **3.Unified Conditional Training:** We merge all realigned datasets into a single mixed set $\mathcal{D}_{\text{unified}} = \mathcal{D}'_1 \cup \mathcal{D}'_2 \cup \dots \cup \mathcal{D}'_N.$
>
> The experimental results of this exploratory method are shown in the table below, demonstrating the feasibility and promise of this multi-target learning:
>
> | Task | $\kappa$ | multi(R) | multi(C) | single(R) | single(C) |
> |---|---|---|---|---|---|
> | **easysparse** | 10 | 0.70 | 0.48 | 0.63 | 0.26 |
> |  | 20 | 0.81 | 1.12 | 0.80 | 1.08 |
> || 40 | 0.86 | 0.70 | 0.89 | 0.81 |
> | **BallCircle** | 10 | 0.68 | 0.90 | 0.65 | 0.81 |
> |  | 20 | 0.71 | 0.76 | 0.70 | 0.79 |
> | | 40 | 0.73 | 0.39 | 0.65 | 0.16 |
> | **AntVelocity** | 10 | 1.00 | 0.73 | 0.98 | 0.45 |
> | | 20 | 1.00 | 0.53 | 0.99 | 0.46 |
> |  | 40 | 0.99 | 0.32 | 1.00 | 0.34 |
>
> For example, in easysparse, BallCircle, and AntVelocity tasks across different κ values, a single 'multi-target' B2R agent achieves comparable R and C performance compared to 'single-target' agents, though multi-target performance often shows slightly higher R and C values. We will discuss the feasibility of this approach in the appendix of the revised paper.
>
> > **W4:** Line 238: The paper states that “these approaches discard potentially useful trajectories once they exceed the limit,” but B2R also performs filtering based on the same cost threshold. Could the authors clarify how B2R’s filtering is fundamentally different or more selective?
>
> We sincerely thank the reviewer for highlighting this confusing sentence. The reviewer is absolutely correct—our original phrasing was imprecise and did not accurately represent our method's mechanism, our intention was to compare how different methods utilize the information within the safe dataset. We apologize for this and will replace the sentence entirely in the revised manuscript.
>
> The fundamental difference lies in the quality and utilization of the supervision signal that each method constructs from the safe trajectories: BC-Safe's signal is sparse and scattered, while TraC's is a coarse, binary classification signal based on partitioning data into "desirable" and "undesirable" sets. In contrast, B2R’s key innovation is to realign all safe trajectories to create a single, dense, and fine-grained supervision signal for the desired constraint.
>
> > **W5:** Appendix C.3: It seems that the cost thresholds for SafetyGymnasium and BulletSafetyGym may have been inadvertently swapped.
>
> We sincerely thank you for your meticulous catch. The values were swapped due to a typo and have been corrected in the revised manuscript. We appreciate your help in improving our paper's accuracy.
>
> > **Q1:** How does B2R perform in datasets with sparse or skewed distributions of safe trajectories, such as under tight cost limits (e.g., FISOR-style setups)?
>
> We understood the reviewer's question as a stress test of B2R's performance under "FISOR-style setups"—challenging scenarios with stringent, low cost-limits that result in a sparse distribution of available safe data.
>
> To address this directly, we conducted a head-to-head comparison against FISOR using its own paper's strict threshold settings ($κ$ =10 for Safety-Gymnasium, $\kappa$ =5 for others). The results are as follows:
>
> |Task|**B2R(R)**|**B2R(C)**|**FISOR(R)**|**FISOR(C)**|
> |---|---|---|---|---|
> |**BallCircle**|**0.60**|**0.98**|0.34|0.00|
> |**DroneCircle**|**0.53**|**0.58**|0.48|0.00|
> |**easysparse**|**0.60**|**0.00**|0.38|0.53|
> |**medmean**|**0.74**|**0.08**|0.39|0.08|
> |**harddense**|**0.42**|**0.14**|0.30|0.34|
> |**AntVelocity**|**0.97**|**0.99**|0.89|0.00|
>
> The conclusion from this experiment is clear: B2R not only remains robustly safe under these stringent conditions but also consistently achieves significantly higher rewards. While FISOR's hard-constraint approach can be overly conservative, B2R learns a more effective policy that better balances safety and performance.

---

> > ### Comment · Reviewer_LsnC · 2025-08-05
> >
> > I appreciate the authors’ efforts in addressing my previous concerns. I have no further questions.

---

### Official Review · Reviewer_yMKb · 2025-07-04

**Clarity:** 3
**Significance:** 2
**Originality:** 2
**Rating:** 5
**Confidence:** 4

**Summary:**

This paper proposes Boundary-to-Region (B2R) Supervision, a framework for offline safe RL using transformer-based sequence models. The key insight is that reward-to-go and cost-to-go should not be treated symmetrically as in CDT, especially when cost-to-go (CTG) corresponds to rigid safety constraints. B2R first filters unsafe trajectories from the offline dataset, then realigns the cost-to-go of all remaining (safe) trajectories to match the safety threshold, enriching the training data for the sequence model. Empirical results show that it achieves better reward while satisfying the cost constraint in most tasks.

**Questions:**

1. Did the author assess “buffered threshold” baseline (CDT trained with a stricter threshold)?

2. How does B2R handle cases where safe trajectories are rare? What is the minimum “safe coverage” required?

3. How does RoPE compare to standard positional embeddings in CDT or Decision Transformer? Any reason to prefer RoPE?

4. Can the authors clarify exactly how region-conditional supervision is used during training and inference? Is it purely boundary-conditioned after cost-to-go shifting?

**Ethical Concerns:**

["NO or VERY MINOR ethics concerns only"]

**Final Justification:**

All major concerns are addressed and authors have provided additional result to strengthen their claims. Final review rating raised to 5.

**Limitations:**

The paper pointed out the limitation of the method in environment with limited safe data in the conclusion.

**Paper Formatting Concerns:**

No major formatting concerns.

**Quality:**

3

**Strengths And Weaknesses:**

### Strengths

* Comprehensive empirical study: 38 tasks from the DSRL benchmark are evaluated, with strong results on both safety and reward metrics. Ablations of key components are included.

* Theoretical analysis: Provide probabilistic guarantees on safety and reward performance of B2R.

* Simple, practical method: Trajectory filtering and cost realignment are easy to implement and does not introduce more hyperparameters. The approach can be integrated into standard Decision Transformer-style methods.

### Weaknesses

1. **Obvious baseline not included**: The policy oscillation around the boundary in Fig2 is to be expected if the model is trained only on near-boundary trajectories. A natural baseline --- CDT trained with a buffer (i.e., a margin below the actual constraint) --- should be included (CDT with a margin), to directly test if realignment is strictly necessary or just a margin effect.

2. **Unclear use of region-conditional supervision**: The role of region-conditional cost supervision (R_t) is not clearly explained. After cost-to-go shifting, is the model still conditioned on a range, or only on the boundary? Is region conditioning used at training or inference, or both?

3. **RoPE rationale and ablation missing**: Rotary position embeddings are used but not discussed in detail. The authors should explain why RoPE helps in this context, and provide comparisons with standard positional embeddings as in Decision Transformer / RvS.

4. **Assumption of sufficient safe data after trajectory filtering**: B2R filters out unsafe trajectories entirely. In domains with limited safe data, this may significantly reduce coverage or reward diversity.

5. **Incremental novelty**: The incremental tweaks --- CTG realignment, trajectory filtering, RoPE --- seems a little bit arbitrary and heuristic. The authors might want to highlight the novel contribution of this paper.

6. **Related work lacks differentiation**: The Related Work section lists prior approaches but does not explicitly position or differentiate B2R from them.  A concise positioning paragraph would help readers and reviewers assess novelty.

7. **Typo in main text**: Line 197 refers to Equation 4.3, which I think should be Equation 9.

---

> ### Author Rebuttal · Authors · 2025-07-30
>
> We sincerely thank reviewer yMKb for your valuable comments and recognition of our work. As per your suggestions, we have responded to each point below and revised the manuscript accordingly.
> >**W1:Obvious baseline not included.** The policy oscillation around the boundary in Fig2 is to be expected if the model is trained only on near-boundary trajectories. A natural baseline --- CDT trained with a buffer (i.e., a margin below the actual constraint) --- should be included (CDT with a margin), to directly test if realignment is strictly necessary or just a margin effect.
>
> >**Q1:** Did the author assess “buffered threshold” baseline (CDT trained with a stricter threshold)?
>
> We thank the reviewer for the invaluable suggestion to benchmark against **CDT with a buffered threshold**. We have conducted the proposed experiment comparing B2R against CDT trained with buffered thresholds (i.e., target CTGs of 70%, 80%, 90%, and 100% of the constraint κ). The results not only highlight B2R's superiority but also expose the fundamental limitations of the buffered baseline approach.
>
> The experimental results are as follows:
>
> |Task|Met|70%|80%|90%|100%|B2R|
> |----|---|--|--|--|---|---|
> |AntVelocity|R|0.95|0.99|0.99|0.99|**0.99**|
> ||C|0.40|0.47|0.53|0.55|**0.42**|
> |CarCircle|R|0.71|0.71|**0.72**|0.72|0.71|
> ||C|0.61|0.70|**0.76**|0.90|0.68|
> |mediummean|R|0.86|0.87|0.82|0.81|**0.88**|
> ||C|1.85|1.97|1.86|1.92|**0.63**|
> |harddense|R|0.19|0.20|0.24|0.25|**0.48**|
> ||C|0.40|0.40|0.55|0.65|**0.58**|
>
> Our key findings are as follows:
>
> - **CDT faces a direct safety-performance conflict**: As the buffered threshold tightens (from 70% to 100% of the constraint value κ), cost decreases but reward is sacrificed, revealing a typical yet unfavorable trade-off.
>
> - **Margin adjustment is unstable and unreliable**: In the medmean task, CDT's margin-cost relationship becomes unstable, and no margin setting satisfies safety constraints, while B2R remains safe (0.63) with higher reward. This shows that for complex tasks, simply tuning the target cost-to-go is a fragile and unreliable solution.
>
> The root cause lies in the learning signal: The "buffered CDT" approach is limited by sparse "boundary supervision"—it only mimics trajectories whose cumulative costs are close to a specific target CTG during inference. In contrast, B2R enables "region supervision" through cost realignment, leveraging all safe trajectories. This yields denser and more diverse training signals, allowing B2R to learn more robust policies and break the inherent limitations of baseline methods.
>
> These results confirm that B2R's core contribution lies in its unique supervision paradigm. We will include this important comparison in the appendix.
>
> > **W2: Unclear use of region-conditional supervision.** The role of region-conditional cost supervision (R_t) is not clearly explained. After cost-to-go shifting, is the model still conditioned on a range, or only on the boundary? Is region conditioning used at training or inference, or both?
>
> >**Q4:** Can the authors clarify exactly how region-conditional supervision is used during training and inference? Is it purely boundary-conditioned after cost-to-go shifting?
>
> We thank the reviewer for this important question, which allows us to clarify the central mechanism of our work.
>
> Region-conditional supervision refers to the training strategy in which the model is supervised by every trajectory whose cost-to-go satisfies the safety constraint, i.e., $C_0 \le \kappa$. This collection of trajectories forms the “safe region” $\mathcal{R}_t$. Unlike conventional approaches that rely only on samples within a narrow margin around the boundary, $C_0\in[\kappa-\varepsilon,\kappa+\varepsilon]$, **region-conditional supervision exploits a broader and more diverse set of feasible behaviors, thereby improving both training stability and generalization.**
>
> To achieve this, we preprocess the data with CTG Realignment: after discarding unsafe trajectories whose cost exceeds the threshold $\kappa$, we apply a constant offset to the CTG sequence of every remaining safe trajectory so that all initial CTGs in the “safe region” are aligned to the constraint boundary $\kappa$. This ensures that **while the CTG input is boundary-conditioned to a fixed $\kappa$, the supervision signal is drawn from the entire safe region.**
>
> **This design unifies training and inference.** At inference, we provide the same initial condition $\kappa$. Because the model has learned to associate this boundary condition with the wide spectrum of safe-region behaviors, **it generates stable and reliable actions that satisfy the constraint, rather than merely imitating a few boundary trajectories.**
>
> > **W3:RoPE rationale and ablation missing.** Rotary position embeddings are used but not discussed in detail. The authors should explain why RoPE helps in this context, and provide comparisons with standard positional embeddings as in Decision Transformer / RvS.
>
> >**Q3:** How does RoPE compare to standard positional embeddings in CDT or Decision Transformer? Any reason to prefer RoPE?
>
> Thank you for pointing this out; we apologize for the lack of clarity in the original manuscript, which led to an important misunderstanding: The “W/O RoPE’’ ablation Section 5.2 and Figure 6 is the requested comparison against the standard absolute positional encoding (APE), and it empirically shows RoPE's superiority. The results clearly show that the RoPE-equipped B2R variant significantly outperforms the APE variant in cost control, empirically demonstrating the superiority of RoPE within our framework.
>
> The choice of RoPE is theoretically grounded: B2R performs **additive CTG realignment**, which demands that the model capture the fine-grained, step-by-step evolution of costs. RoPE’s relative positional encoding **offers a strong inductive bias for learning these local, step-level dynamics**, enabling the model to better understand the relative causal relationships between actions and cost changes. By contrast, APE provides only absolute positional information and is less suited to modeling such relative dynamics.
>
> We will update the label and add a more detailed theoretical explanation to the paper.
>
> >**W4:Assumption of sufficient safe data after trajectory filtering.** B2R filters out unsafe trajectories entirely. In domains with limited safe data, this may significantly reduce coverage or reward diversity.
>
> >**Q2:** How does B2R handle cases where safe trajectories are rare? What is the minimum “safe coverage” required?
>
> We thank the reviewer for this critical question regarding performance with sparse safe data. This is a crucial consideration for any practical offline safe RL method, and we agree it represents a fundamental challenge for the field.
>
> To address this, we conducted a new few-shot experiment. Due to character limits, we respectfully refer to our detailed response to Reviewer LsnC (W1) for the full results. The results show that while performance is naturally affected by less data, B2R exhibits remarkable stability and a graceful degradation profile. Our region-wide supervision is designed to be highly data-efficient, as it extracts a learning signal from every single available safe trajectory.
>
> **Regarding a "minimum safe coverage," this experiment shows it is highly task-dependent.** For extreme cases, we believe future work could further enhance B2R's performance by integrating data augmentation or generative modeling techniques to create more diverse safe samples.
>
> >**W5:The incremental tweaks.** --- CTG realignment, trajectory filtering, RoPE --- seems a little bit arbitrary and heuristic. The authors might want to highlight the novel contribution of this paper.
>
> We agree that when viewed in isolation, trajectory filtering, CTG realignment, and RoPE may appear as incremental tweaks. However, we wish to emphasize that their combination is not an ad-hoc assembly but a deliberately engineered, interlocking whole designed to instantiate a fundamentally new paradigm.
>
> The true novelty of our work lies in identifying the symmetry fallacy inherent in prior methods—the failure to recognize the asymmetry between the flexibility of rewards and the rigidity of costs, which hinders effective learning of the safety boundary.
>
> To resolve this foundational issue, we introduce a region-level supervision paradigm that conditions the policy on the cost-to-go distribution of the entire safe region rather than on sparse boundary samples. Implementing this paradigm demands a coherent pipeline:
>
> 1. **Trajectory filtering** is the prerequisite that **defines the safe region**. Without this step, the concept of learning from the whole region is moot.
> 2. **CTG realignment** is the **core mechanism** that converts heterogeneous safe trajectories into a **uniform, dense supervision signal** anchored at the single safety boundary $\kappa$.
> 3. **RoPE** is the **synergistic component** that preserves the temporal fidelity of this refined signal. As detailed in our responses to W3 & Q3, RoPE provides the inductive bias required to model the incremental cost dynamics introduced by realignment.
>
> Together, these three components form a **logically coherent and mutually reinforcing system** purpose-built from the ground up to realize our central idea. Empirical results robustly validate the efficacy of this system in achieving the proposed paradigm.
>
> >**W6:Related work lacks differentiation.** The Related Work section lists prior approaches but does not explicitly position or differentiate B2R from them. A concise positioning paragraph would help readers and reviewers assess novelty.
>
> We agree B2R's positioning needed strengthening. Per your suggestion, we will add a positioning paragraph in revised paper to explicitly differentiate B2R from key methods like CDT and TraC to highlight its novelty.
>
> >**W7:Typo in Line 197.**
>
> We fixed this typo in our revised version and carefully checked the manuscript.

---

> > ### Comment · Reviewer_yMKb · 2025-08-06
> >
> > I thank the authors for conducting the additional experiments (comparing against CDT with buffered threshold & safe trajectory sufficiency analysis). I have no other questions.

---

### Author Response · Authors · 2025-08-06
**Final Thanks to All Reviewers and Chairs**

Dear Reviewers, Area Chair, and Senior Area Chair,

Thank you all for your time and insightful feedback throughout the review and discussion process. We are very grateful for your engagement and the positive final comments affirming that our rebuttal has successfully addressed your concerns.

We will be sure to incorporate all the valuable feedback—including the new experimental results and the helpful paper reference—into the final version of our paper.

We sincerely appreciate your selfless contributions to the community.

Thank you once again.

Best regards,
The Authors

---

### Decision · Program_Chairs · 2025-09-17

**Decision:**

Accept (poster)

**Comment:**

This paper considers the problem of offline safe reinforcement learning (OSRL) where the goal is to learn a reward maximizing policy with respect to a given cost constraint from a given offline dataset. Specifically, it identifies a fundamental limitation of sequence modeling based OSRL methods (e.g., decision transformer): they treat return-to-go (RTG) and cost-to-go (CTG) equivalently. The paper argues why this is not appropriate because RTG is a flexible objective to maximize whereas CTG corresponds to a strict safety budget constraint -- leads to unreliable safety, especially for OOD cost trajectories. The paper proposes a intuitive approach referred to as Boundary-to-Region (B2R) to address this limitation by enabling asymmetric conditioning of RTG and CTG via trajectory filtering, CTG realignment, and rotary positional embeddings. Under simplifying assumptions, B2R is analyzed theoretically. Experiments on 38 benchmark tasks demonstrate the efficacy of B2R over prior baseline methods.

All the reviewers' appreciated the paper's contributions but also raised several good questions. The author response addressed these questions adequately resulting in increase of scores in some cases. Overall, this is a good paper on an important topic. I recommend accepting the paper and strongly encourage the authors' to incorporate all the discussion into the final paper and also release the code.